# MoE-Pruner: Pruning Mixture-of-Experts Large Language Model using the Hints from Its Router

## Abstract

Mixture-of-Experts (MoE) architectures face challenges such as high memory consumption and redundancy in experts. Pruning MoE can reduce network weights while maintaining model performance. Motivated by the recent observation of emergent large magnitude features in Large Language Models (LLM) and MoE routing policy, we propose MoE-Pruner, a method that prunes weights with the smallest magnitudes multiplied by the corresponding input activations and router weights, on each output neuron. Our pruning method is one-shot, requiring no retraining or weight updates. We evaluate our method on Mixtral-8x7B and Mixtral-8x22B across multiple language benchmarks. Experimental results show that our pruning method significantly outperforms state-of-the-art LLM pruning methods. Furthermore, our pruned MoE models can benefit from a pretrained teacher model through expert-wise knowledge distillation, improving performance post-pruning. Experimental results demonstrate that the Mixtral-8x7B model with 50% sparsity maintains 99% of the performance of the original model after the expert-wise knowledge distillation.

## 1 Introduction

Scaling neural network models is one of the main drivers of better performance in deep learning. From BERT (Devlin et al., 2019) to GPT-3 (Brown et al., 2020) to Llama 3.1 405B (Dubey et al., 2024) in natural language processing, or from ResNet (He et al., 2016) to ViT (Dosovitskiy et al., 2021) in computer vision, breakthroughs in performance have been obtained from larger models, datasets, and computational resources for training (Kaplan et al., 2020). However, the cost of training state-of-the-art models grows exponentially. For instance, BERT-Large (345M parameters, proposed in 2018) requires an estimated $5 \times 10^{20}$ FLOPs (Devlin et al., 2019) to train, GPT-3 (175B parameters, from 2020) requires $3.14 \times 10^{23}$ FLOPs (Brown et al., 2020), while Llama 3.1 (405B, released in 2024) requires $3.8 \times 10^{25}$ FLOPs (Dubey et al., 2024) to train. This exponential growth motivates researchers to seek more efficient and effective training approaches.

Mixture-of-Experts (MoE) architectures (Jacobs et al., 1991; Shazeer et al., 2017) have been proposed to reduce the computing cost while enabling efficient scaling of network capacity. It has been successfully employed to scale both vision (Ruiz et al., 2021; Shen et al., 2023) and language (Lepikhin et al., 2021; Fedus et al., 2022) models. In addition, these models provide other advantages, including sparsity that can mitigate catastrophic forgetting in continual learning and an inductive bias that can enhance performance in multitask learning (Collier et al., 2020; Komatsuzaki et al., 2023). Overall, MoE has proven to be a promising strategy for scaling deep learning models across various domains.

However, several crucial limitations persist in MoE for expanding its capacity. First of all, the static parameters, particularly those required for constructing the MoE architecture, introduce substantial memory overheads and constraints for deployment. For example, Mixtral-8x7B (Jiang et al., 2024) expert layers account for 96% of model parameters (45B out of 47B), which demands considerable memory and storage during inference. Moreover, MoE has a poor utilization of its experts. The conventional learning-based routing policy for MoE suffers from representation collapse issues since

it encourages token embeddings to be clustered around expert centroids (Chi et al., 2022) and results in redundant experts (Mittal et al., 2022; Chen et al., 2022).

One possible solution to address those drawbacks and fully unleash the power of MoE is consolidating information from insignificant experts, aiming to establish a more compact MoE without hurting performance. Another solution is pruning experts that yield the lowest token reconstruction loss. Nevertheless, naively combining existing model merging mechanisms or expert pruning leads to performance degradation in the MoE architectures. We raise the following pivotal questions for MoE LLM pruning: (i) How do we formulate and devise comprehensive pruning metrics that leverage existing methods? (ii) How do we find the optimal pruning metric tailored for MoE Large Language Models?

In this paper, we systematically explore MoE LLM pruning and target a high-quality compressed MoE model in downstream fine-tuning scenarios. Specifically, we first analyze the open-source MoE model's expert activation frequency and observe that different MoE expert initialization methods result in different expert activation frequencies and expert similarities. We leverage existing LLM pruning methods such as SparseGPT (Frantar & Alistarh, 2023b) and Wanda (Sun et al., 2024), and design a novel pruning metric that incorporates MoE router weights information to identify and remove unimportant weights in expert layers. Since the pruning process is one-shot and only requires a small set of calibration data, the MoE model suffers from performance degradation. To recover MoE model performance, we further propose an expert-wise knowledge distillation method that utilizes the pretrained model as a teacher model, facilitating the recovery of the pruned model's performance.

Our main contributions can be summarized as follows:

- We propose a novel framework, MoE-Pruner, that is efficient and effective for pruning MoE models with minimal performance degradation.
- We design an innovative expert-wise knowledge distillation method that leverages the pretrained MoE model as a teacher model to recover pruned MoE student model performance.
- Experimental results on Mixtral MoE models across nine zero-shot evaluation benchmarks demonstrate the effectiveness of our MoE-Pruner algorithm. MoE-Pruner achieves minimal performance drop even at 50% sparsity with only a small set of calibration data compared with existing pruning methods. The pruned model maintains 99% of the performance of the original model after the expert-wise knowledge distillation.

## 2 PRELIMINARIES

**Mixture-of-Experts (MoE).** Scaling model size increases learning capacity and enhances generalization (Kaplan et al., 2020; Brown et al., 2020; Hoffmann et al., 2022). MoE (Jacobs et al., 1991; Shazeer et al., 2017; Lepikhin et al., 2021; Fedus et al., 2022) is an efficient approach that enables significantly more compute-efficient pretraining and inference. It replaces the feed-forward network (FFN) layers in Transformers (Vaswani et al., 2017) with expert layers, where different experts are activated for different input tokens instead of utilizing the full network parameters. Sparse MoE architecture can dramatically scale the model with the same compute budget as a dense model.

**MoE Expert Initialization.** MoE expert initialization uses different strategies, which can be classified into two categories: **sparse upcycling** (Komatsuzaki et al., 2023) and **training from scratch**. The sparse upcycling method starts from a dense model checkpoint and copies all parameters, except the MoE router, which does not exist in the original dense model. In particular, each expert in the new MoE layer is an identical copy of the original MLP layer that is replaced. Some open-source MoE models such as Mixtral (Jiang et al., 2024), Qwen1.5-MoE-A2.7B (Team, 2024), and MiniCPM-MoE (Hu et al., 2024) all employ the upcycling approach to reduce the total training costs. While some MoE models like DeepSeek-V2 (Liu et al., 2024), OLMoE (Muennighoff et al., 2024), and Yuan2.0-M32 (Wu et al., 2024) use the training from scratch approach to help expert diversification.

**Large Language Model Pruning.** Magnitude pruning (Han et al., 2016) is a standard approach to induce sparsity in neural networks. It removes individual weights with magnitudes below a certain threshold. However, magnitude pruning fails dramatically on LLMs even with relatively low levels

of sparsity (Frantar & Alistarh, 2023b). SparseGPT (Frantar & Alistarh, 2023b) proposes a one-shot, post-training pruning method that prunes LLM weights and uses Hessian matrix and calibration data to update the remaining weights without any retraining. Wanda (Sun et al., 2024) is a simple method that prunes LLM weights with the smallest magnitudes multiplied by the corresponding input activations without any additional weight update.

## 3 METHODOLOGY

### 3.1 THE MIXTURE-OF-EXPERTS ARCHITECTURE

**Mixture-of-Experts (MoE) architecture.** MoE architecture replaces the feed-forward networks (FFN) in Transformers with mixture-of-expert layers. A router or a gating network is trained to select a subset of experts for each input token based on its routing policy. Given $n$ experts in a layer, the output of the expert layer is given by:

$$y = \sum_{i=0}^{n-1} Gate(x)_i \cdot E_i(x), \tag{1}$$

where the $Gate(x)_i$ is the router weights from the gating network assigned to the $i$-th expert, and $E_i(x)$ is the output of $i$-th expert. The router weights can be formulated as softmax over the Top-K logits:

$$Gate(x) = \text{Softmax}(\text{TopK}(x \cdot W_g)), \tag{2}$$

where $W_g$ is the weight of the router or gating network, and $\text{TopK}(X)_i = l_i$ if $i$ is in the top-K coordinates of logits $l$ and $\text{TopK}(X)_i = -\infty$ otherwise.

Since current LLMs mostly adopt SwiGLU (Shazeer, 2020) architecture for the FFN, and MoE LLM such as Mixtral-8x7B (Jiang et al., 2024) uses a top-2 to select experts, we can derive the output of an expert layer as:

$$y = \sum_{i=0}^{n-1} \text{Softmax}(\text{Top2}(x \cdot W_g))_i \cdot \text{SwiGLU}_i(x). \tag{3}$$

Some recent MoE LLMs, such as DeepSeekMoE (Dai et al., 2024), adopt shared experts that are always activated, aiming at capturing and consolidating common knowledge across varying contexts.

**MoE Expert Activation Frequency.** We use a subset of the C4 (Raffel et al., 2020) dataset and collect the activation frequency of MoE experts. The expert activation frequency is task-agnostic since C4 pretraining datasets are comprehensive and not dominated by knowledge specific to any particular domain. Motivated by the load balancing loss (Shazeer et al., 2017; Lepikhin et al., 2021; Fedus et al., 2022), we propose to use the coefficient of variation of expert activation frequency in each layer to represent the load balancing score, where a lower score represents more balanced loads. Given $n$ experts and $l$ layers and a batch $\mathcal{B}$ with $T$ tokens, the load balancing score for one layer is:

$$s = \frac{\sigma}{\mu} = \frac{\sqrt{\frac{1}{n} \sum_{i=0}^{n-1} (f_i - \mu)^2}}{\mu},$$

$$\mu = \frac{1}{n} \sum_{i=0}^{n-1} f_i, \tag{4}$$

where $f_i$ is the number of tokens dispatched to expert $i$,

$$f_i = \sum_{x \in \mathcal{B}} \mathbb{1}\{\text{argmax } p(x) = i\}. \tag{5}$$

We can derive the load balancing score by calculating the mean of scores across all $l$ MoE layers, such that we can use this score to compare with various MoE models with different numbers of experts.

Figure 1 shows the load balancing scores of Mixtral-8x7B (Jiang et al., 2024), Qwen-1.5-A2.7B (Team, 2024), DeepSeek-V2 and DeepSeeek-V2-Lite (Liu et al., 2024), MiniCPM-MoE-8x2B (Hu et al., 2024), and OLMoE (Muennighoff et al., 2024). We find that different MoE expert

initialization methods result in different expert activation frequencies and expert similarities, which will impact the MoE pruning strategies. For instance, the MoE model initialized with upcycling can take advantage of the dense model and reduce training costs. The final MoE model exhibits higher expert similarity and more balanced expert activation frequency, which indicates that expert pruning will result in a performance drop, and weight pruning will be a better choice. MoE model trained from scratch might yield better performance as it avoids the limitations of starting with a group of identical experts, which can hinder diversification (Wei et al., 2024). It also shows imbalanced expert activation frequency, indicating that least-used expert pruning could help compress model size and not bring performance degradation for task-dependent setting.

## 3.2 PRUNING METRIC

**Problem Formulation.** Post-training pruning for LLMs can be decomposed into layer-wise subproblems (Lu et al., 2022; Frantar & Alistarh, 2023b; Sun et al., 2024; Dong et al., 2024). Given a sparsity ratio and a linear layer with weight $\mathbf{W}$, the pruning algorithm tries to find a sparsity mask $\mathbf{M}$ that minimizes reconstruction loss:

$$\underset{\mathbf{M}}{\mathrm{argmin}}\|\mathbf{W}\mathbf{X} - (\mathbf{M} \odot \mathbf{W})\mathbf{X}\|. \quad (6)$$

Optimal Brain Damage (OBD) (LeCun et al., 1989) first sets up a pioneering framework for neural network pruning. It uses second-order information without off-diagonal elements in the Hessian matrix for faster approximation. Optimal Brain Surgeon (OBS) (Hassibi et al., 1993) develops upon OBD partly by taking into account the off-diagonal elements. SparseGPT (Frantar & Alistarh, 2023b) revisits the OBS, computes the inverse Hessian only once, and reuses to update weight in the remaining rows that are also in the mask to mitigate reconstruction loss. The pruning metric in SparseGPT is:

$$\mathcal{S}_{ij} = [|\mathbf{W}|^2/\mathrm{diag}(\mathbf{H}^{-1})]_{ij}. \quad (7)$$

Wanda (Sun et al., 2024) further simplifies the pruning metric to the following form without the need to compute the inverse of the Hessian matrix $\mathbf{H}$:

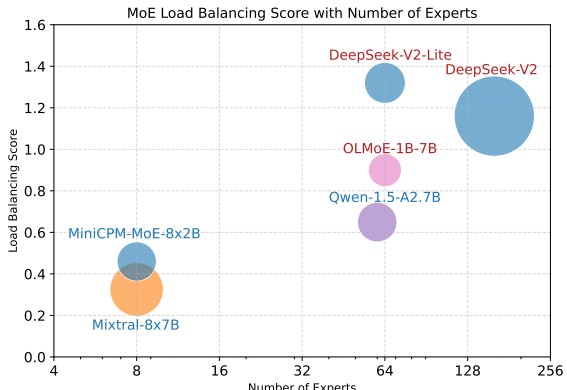

Figure 1: Load balancing score of MoE models. We collect the expert activation frequency of MoE models and calculate the load balancing score (lower is better). The circle area represents the model size. MoE models trained from scratch are marked with red, while MoE models that use upcycling are marked with blue. MoE models trained from scratch usually have more experts and imbalanced loads. MoE models initialized with upcycling tend to have more balanced loads and less number of experts. The only exception is Qwen-1.5-A2.7B, which is initialized with upcycling. But according to the report (Yang et al., 2024), its expert parameters are shuffled along the intermediate dimension to guarantee that each fine-grained expert exhibits unique characteristics and therefore exhibits more like trained from scratch MoE models.

$$\mathcal{S}_{ij} = [|\mathbf{W}|^2/\mathrm{diag}((\mathbf{X}^T\mathbf{X})^{-1})]_{ij} \approx [|\mathbf{W}|^2/(\mathrm{diag}(\mathbf{X}^T\mathbf{X})^{-1})]_{ij} = (|\mathbf{W}_{ij}| \cdot \|\mathbf{X}_j\|)^2. \quad (8)$$

When it comes to pruning MoE, the expert layers constitute the majority of model parameters. For example, the Mixtral-8x7B (Jiang et al., 2024) has a total of 47B parameters where 1.3B belongs to attention modules and 45B is used for expert layers (2 out of 8 experts are activated, 12.5B active parameters during inference). Only a subset of experts are activated for different input tokens, so there is a large space of expert redundancy.

**Motivation.** Consider a simple Mixture-of-Experts with two experts and each with only one weight: $y = Gate(x)_1 \cdot E_1(x) + Gate(x)_2 \cdot E_2(x) = Gate_1 \cdot w_1 \cdot x + Gate_2 \cdot w_2 \cdot x$, where $|w_1| \le |w_2|$. If we want to remove one weight without incurring significant change on the output, traditional magnitude pruning (Han et al., 2016) will remove weight $|w_1|$. However, in MoE architecture, the router weights $Gate_i$ is an important part as it assigns different values to different experts. Especially

when we consider a top-k setting that only a subset of experts are activated, the router weights $Gate_1$ could be a large value close to 1, while router weights $Gate_2$ could be 0 if it is not activated. As a results, $|Gate_1 \cdot w_1 \cdot x| \gg |Gate_2 \cdot w_2 \cdot x|$, and therefore we should remove weight $w_2$ instead to minimize change on the output.

This motivating example shows that for MoE architecture, we need to consider the importance of router weights. Previous pruning methods for LLMs do not consider the router weights which only exist in MoE architecture and may result in lower performance after pruning MoE. We propose a pruning metric designed explicitly for MoE LLMs to handle such a limitation while maintaining the simplicity of Wanda's pruning metric.

**Router Tells It All.** Motivated by the pruning metric in Wanda and the MoE routing policy, our approach, MoE-Pruner, prunes weights with the smallest magnitudes multiplied by the corresponding input activations and router weights, on each output neuron:

$$\mathcal{S}_{ij} = |\mathbf{W}_{ij}| \cdot \|\mathbf{X}_j \cdot \mathbf{Gate}\|, \tag{9}$$

where $i$ and $j$ stands for output feature and input feature dimension, respectively.

Table 1: Comparison of different pruning methods including magnitude pruning, SparseGPT, Wanda, and MoE-Pruner.

| Method | Weight update | Calibration data | Pruning metric $\mathcal{S}_{ij}$ | Complexity |
|---|---|---|---|---|
| Magnitude | ✗ | ✗ | $|\mathbf{W}|$ | $\mathcal{O}(1)$ |
| SparseGPT | ✔ | ✔ | $[|\mathbf{W}|^2/\mathrm{diag}(\mathbf{H}^{-1})]_{ij}$ | $\mathcal{O}(d_{hidden}^3)$ |
| Wanda | ✗ | ✔ | $|\mathbf{W}_{ij}| \cdot \|\mathbf{X}_j\|$ | $\mathcal{O}(d_{hidden}^2)$ |
| MoE-Pruner | ✗ | ✔ | $|\mathbf{W}_{ij}| \cdot \|\mathbf{X}_j \cdot \mathbf{Gate}\|$ | $\mathcal{O}(d_{hidden}^2)$ |

Table 1 summarizes pruning methods, including magnitude pruning, SparseGPT, Wanda, and MoE-Pruner and their corresponding pruning metric and complexity. Algorithm 1 presents the unstructured sparsity version of our MoE-Pruner algorithm. Our method is simple and efficient for MoE models and does not require a sophisticated weight update procedure.

---

**Algorithm 1** The MoE-Pruner algorithm. We prune each expert layer weight matrix $\mathbf{W}$ to $p\%$ sparsity.

---

1: **Initialize:** A MoE model $\mathcal{M}$ with $l$ MoE layers, where each MoE layer has $n$ experts. Let $\mathbf{X} \in \mathbb{R}^{b \times d_{\mathrm{col}}}$ and $\mathbf{Gate} \in \mathbb{R}^{b \times n}$ denote the *calibration samples* and *router weights* respectively.
2: **for** layer $t = 1, \ldots, l$ **do**
3:     $\mathbf{X}', \mathbf{Gate} \leftarrow \texttt{forward}(\mathrm{layer}_t, \mathbf{X})$      ▷ get router weights
4:     **for** expert $e = 1, \ldots, n$ **do**
5:         $\mathbf{M} \leftarrow \mathbf{1}_{d_{\mathrm{row}} \times d_{\mathrm{col}}}$      ▷ binary pruning mask
6:         $\mathcal{S}_{ij} \leftarrow |\mathbf{W}_{ij}| \cdot \|\mathbf{X}_j \cdot \mathbf{Gate}\|$      ▷ compute pruning metric
7:         $idx \leftarrow \texttt{sort}(\mathcal{S}_{ij}, dim = 1)$      ▷ prune weights indices based on metric
8:         $\mathbf{M} \leftarrow \texttt{scatter}(0, idx_{:, d_{\mathrm{col}} * p\%})$
9:         $\mathbf{W} \leftarrow \mathbf{M} \odot \mathbf{W}$      ▷ set pruned weights to 0
10:     **end for**
11:     $\mathbf{X} \leftarrow \mathbf{X}'$
12: **end for**
13: **Return:** A pruned MoE model $\mathcal{M}'$.

---

**Structured N:M Sparsity**. Structured N:M sparsity can leverage NVIDIA's sparse tensor cores to accelerate matrix multiplication. While MoE-Pruner so far has been developed for unstructured sparsity, it can be easily extended to structured N:M sparsity (Mishra et al., 2021), where we compare weights using the same metric among every M consecutive weights, for all weights connected to an output.

**Comparison Group**. Generally, for a pruning method, each weight is first assigned an importance score, calculated by the pruning metric. These weights are then grouped into comparison groups

where weights within each group are compared against one another, and weights with lower importance scores are pruned. Most previous pruning methods default to comparing weights locally within each layer or globally across the whole network. Our method compares weights using the comparison groups on each output neuron, which aligns with Wanda and can be easily extended to N:M semi-structured sparsity. Moreover, our method is not limited to pruning individual weights but can also group structured weights into a unit and compare those units among a larger comparison group, such that we can extend our method to structured sparsity.

### 3.3 EXPERT-WISE KNOWLEDGE DISTILLATION

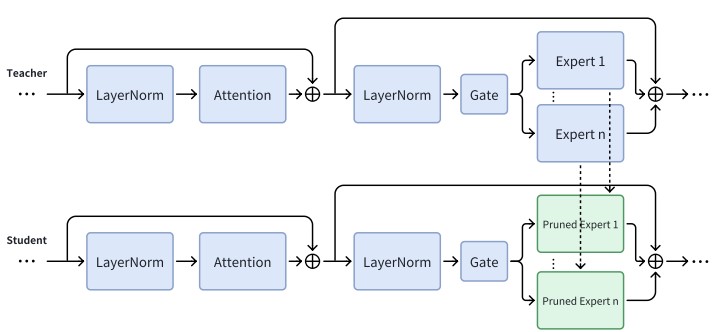

Figure 2: Expert-wise knowledge distillation for the pruned MoE model using the pretrained MoE model as the teacher to recover the performance of the pruned model.

**Expert-Wise Knowledge Distillation.** MoE models can preserve most of their capacity after pruning but still suffer from performance degradation. To recover MoE LLM performance, we fine-tune the model by leveraging the unpruned pretrained model as a teacher model in an expert-wise knowledge distillation (KD) manner. The pretrained model is a natural teacher model for the pruned model since they share exactly the same number of layers, experts, and dimensions (Kurtic et al., 2023). The loss function for expert-wise knowledge distillation is formulated as follows:

$$\mathcal{L}_{KD} = \mathcal{L}_{CE} + \lambda \times \mathcal{L}_{expert} = \mathcal{L}_{CE} + \lambda \times \sum_{j=0}^{l-1} \sum_{i=0}^{n-1} \text{MSE}(E_{it}^j, E_{is}^j), \tag{10}$$

where $\mathcal{L}_{CE}$ is the cross entropy loss, MSE is the mean squared error calculated as $\text{MSE}(X, Y) = \frac{1}{N} \sum_{i=0}^{N-1} (x_i - y_i)^2$ for $N$-dimensional vectors $X$ and $Y$. $\lambda$ is a weighting coefficient and initialized based on the strength of cross entropy loss and expert-wise knowledge distillation loss: $\frac{\mathcal{L}_{CE}}{\mathcal{L}_{expert}}$. We sum up all the differences between teacher experts and student experts. Figure 2 illustrates the expert-wise knowledge distillation for pruned models. The corresponding expert in the pretrained teacher model will be used to distill the expert in the pruned student model.

## 4 EXPERIMENTS

**Models, Datasets, and Evaluation.** We conduct pruning experiments on widely adopted open-source MoE models: the base and instruct version of Mixtral-8x7B and Mixtral-8x22B (Jiang et al., 2024). We use samples from the pretraining dataset C4 (Raffel et al., 2020) as calibration data for one-shot pruning since pretraining datasets are often more comprehensive and not dominated by knowledge specific to any particular domain. We use the exact same 128 sequences of calibration data for all one-shot pruning experiments to control this variable factor. We evaluate the perplexity on the WikiText (Merity et al., 2017) validation set. Our expert-wise knowledge distillation method uses a subset of the C4 (Raffel et al., 2020) as the training set. We measure the performance of pruned models on zero-shot tasks and language modeling. For zero-shot evaluation, we use nine popular tasks from EleutherAI LM Harness (Gao et al., 2023). The nine evaluated zero-shot tasks are: ARC-easy, ARC-challenge (Clark et al., 2018), Boolq (Clark et al., 2019), HellaSwag (Zellers et al., 2019), MMLU (Hendrycks et al., 2021), OpenBookQA (OBQA) (Mihaylov et al., 2018), PIQA (Bisk et al., 2020), RTE (Wang et al., 2018), and WinoGrande (Sakaguchi et al., 2021).

**Baselines and Experiments Setup.** We compare MoE-Pruner with prior pruning approaches, including SparseGPT (Frantar & Alistarh, 2023b) and Wanda (Sun et al., 2024). Similarly, our pruning algorithm is implemented in a layer-wise reconstruction manner. All pruning experiments are conducted on a single NVIDIA H100-80GB GPU. The fine-tuning experiments use the pruned model as a starting point and perform full-parameter fine-tuning to preserve the sparsity mask. We implement the expert-wise knowledge distillation method in Llama-Factory (Zheng et al., 2024) and conduct experiments on 2 servers, each with 8 NVIDIA H100-80GB GPUs. We fine-tune the pruned student model for three epochs, using a learning rate of 2e-5 with the cosine learning rate scheduler.

## 4.1 ONE-SHOT PRUNING

Table 2: WikiText Perplexity against other one-shot pruning methods, including SparseGPT, Wanda, and MoE-Pruner.

| Model | Method | Sparsity | Perplexity ↓ | Sparsity | Perplexity ↓ |
|-------|--------|----------|--------------|----------|--------------|
| Mixtral-8x7B | Pretrained | - | 3.84 | - | 3.84 |
| | SparseGPT | 50% | 4.99 | 40% | 4.40 |
| | Wanda | 50% | 4.97 | 40% | 4.33 |
| | MoE-Pruner (Ours) | 50% | **4.68** | 40% | **4.20** |
| Mixtral-8x7B | Pretrained | - | 3.84 | - | 3.84 |
| | SparseGPT | 2:4 | 7.09 | 1:4 | 4.28 |
| | Wanda | 2:4 | 6.98 | 1:4 | 4.25 |
| | MoE-Pruner (Ours) | 2:4 | **5.88** | 1:4 | **4.14** |
| Mixtral-8x7B-Instruct | Pretrained | - | 4.14 | - | 4.14 |
| | SparseGPT | 50% | 5.20 | 40% | 4.60 |
| | Wanda | 50% | 5.16 | 40% | 4.57 |
| | MoE-Pruner (Ours) | 50% | **4.94** | 40% | **4.48** |
| Mixtral-8x7B-Instruct | Pretrained | - | 4.14 | - | 4.14 |
| | SparseGPT | 2:4 | 7.19 | 1:4 | 4.51 |
| | Wanda | 2:4 | 6.92 | 1:4 | 4.49 |
| | MoE-Pruner (Ours) | 2:4 | **6.11** | 1:4 | **4.41** |
| Mixtral-8x22B | Pretrained | - | 2.83 | | |
| | SparseGPT | 50% | 4.19 | | |
| | Wanda | 50% | 3.97 | | |
| | MoE-Pruner (Ours) | 50% | **3.64** | | |
| Mixtral-8x22B-Instruct | Pretrained | - | 2.89 | | |
| | SparseGPT | 50% | 4.27 | | |
| | Wanda | 50% | 4.06 | | |
| | MoE-Pruner (Ours) | 50% | **3.72** | | |

Table 2 shows the one-shot pruning model perplexity on WikiText with 50% sparsity. There is a clear difference between MoE-Pruner and other pruning methods, including SparseGPT (Frantar & Alistarh, 2023b) and Wanda (Sun et al., 2024). For Mixtral-8x7B (Jiang et al., 2024) models, MoE-Pruner achieves 0.22-0.31 better perplexity over SparseGPT and Wanda. This improvement expands when the MoE model scales to the Mixtral-8x22B model. For the larger Mixtral-8x22B model, MoE-Pruner achieves 0.55 better perplexity over SparseGPT and 0.31-0.34 better perplexity over Wanda. MoE-Pruner further expands the improvement to 1.21 better perplexity over SparseGPT and 1.10 better perplexity over Wanda when we prune the MoE models with the 2:4 semi-structured sparsity.

Table 4 shows the average zero-shot accuracies on nine zero-shot tasks of the pruned Mixtral-8x7B MoE models with 50% unstructured sparsity. The average performance of pretrained models, SparseGPT, Wanda, and our pruned models are 69.16, 66.27, 65.90, and 67.23, respectively. MoE-Pruner outperforms the state-of-the-art pruning approaches, SparseGPT and Wanda, by a large margin. Given that no fine-tuning takes place at this time, there is a noticeable gap between the sparse pruned MoE model and the original pretrained MoE model.

## 4.2 EXPERT-WISE KNOWLEDGE DISTILLATION PERFORMANCE

Table 3: Comparison with NAEE (Lu et al., 2024) about memory reduction and inference speedup on A100. Our proposed MoE-Pruner method at the structured 2:4 sparsity pattern outperforms NAEE in terms of both average performance and inference speedup, while incurring only a small memory overhead for storing sparse tensor indices. The original average performance and memory of Mixtral-8x7B are 69.16 and 87.49GB, respectively.

| Model | Method | Sparsity | Average | Memory(GB) | Speedup |
|-------|--------|----------|---------|------------|---------|
| Mixtral-8x7B | NAEE | r=4 | 61.70 | **45.49** | 1.06× |
| | MoE-Pruner | 2:4 | **64.58** | 50.74 | **1.14×** |

The gap between the pruned MoE model and the pretrained MoE model can be largely mitigated via expert-wise knowledge distillation. We only need 1000 training samples from C4 (Raffel et al., 2020), and training can be done in 1 hour. Table 6 shows the average zero-shot accuracy of the pruned and

Table 4: Average zero-shot performance on 9 evaluation tasks of pruned models using SparseGPT, Wanda, and MoE-Pruner, with 50% unstructured sparsity.

| Model | Method | ARC-c | ARC-e | Boolq | HellaSwag | MMLU | OBQA | PIQA | RTE | WinoGrande | Average |
|-------|--------|-------|-------|-------|-----------|------|------|------|-----|------------|---------|
| Mixtral-8x7B | Pretrained | 56.91 | 84.47 | 85.29 | 64.78 | 67.03 | 35.0 | 82.43 | 70.4 | 76.16 | 69.16 |
| | SparseGPT | 50.43 | 80.68 | 84.62 | 60.20 | 61.79 | 32.8 | 81.12 | **68.59** | **76.16** | 66.27 |
| | Wanda | 51.02 | 80.89 | 85.08 | 60.45 | 62.73 | 32.6 | 80.90 | 64.64 | 74.82 | 65.90 |
| | MoE-Pruner | **53.33** | **81.86** | **86.02** | **62.29** | **64.76** | **33.6** | 81.61 | 66.06 | 75.53 | **67.23** |
| MiniCPM-8x2B | Pretrained | 42.75 | 76.22 | 77.28 | 56.49 | 52.63 | 29.0 | 77.48 | 75.81 | 66.61 | 61.58 |
| | SparseGPT | 39.25 | 73.44 | **76.36** | 53.19 | 48.35 | **28.0** | 76.22 | 64.62 | **64.96** | 58.26 |
| | Wanda | 40.44 | 72.73 | 74.71 | 51.70 | 45.78 | 25.8 | 76.06 | 71.84 | 61.48 | 57.84 |
| | MoE-Pruner | **40.87** | **74.92** | 74.74 | **54.59** | **48.89** | **28.0** | **76.61** | **72.56** | 64.56 | **59.53** |
| DeepSeek-V2-Lite | Pretrained | 46.67 | 78.28 | 79.88 | 58.65 | 54.94 | 34.2 | 80.03 | 61.37 | 71.35 | 62.81 |
| | SparseGPT | 40.36 | 73.70 | 73.27 | 50.37 | 39.85 | 29.0 | 76.66 | 58.12 | 67.25 | 56.51 |
| | Wanda | 41.64 | 73.44 | 71.83 | 51.36 | 39.83 | 29.0 | 77.53 | 63.90 | 66.93 | 57.27 |
| | MoE-Pruner | **44.62** | **76.30** | **78.56** | **55.92** | **49.72** | **31.2** | **78.62** | 60.29 | **70.32** | **60.62** |
| Qwen1.5-MoE-A2.7B | Pretrained | 41.81 | 73.32 | 79.88 | 57.98 | 61.29 | 30.0 | 80.09 | 69.31 | 68.98 | 62.58 |
| | SparseGPT | 34.81 | 68.90 | 76.24 | 49.86 | 51.55 | 25.2 | 77.09 | 55.96 | 67.32 | 56.33 |
| | Wanda | 33.02 | 67.30 | 75.11 | 48.26 | 50.35 | 26.8 | 75.35 | 62.09 | 65.82 | 56.01 |
| | MoE-Pruner | **39.68** | **72.60** | **78.44** | **54.88** | **57.63** | **30.4** | **78.73** | **72.92** | **66.93** | **61.36** |

Table 5: Average zero-shot performance on 9 evaluation tasks of pruned models using SparseGPT, Wanda, NAEE, and MoE-Pruner, at the structured 2:4 sparsity or 50% expert pruning.

| Model | Method | ARC-c | ARC-e | Boolq | HellaSwag | MMLU | OBQA | PIQA | RTE | WinoGrande | Average |
|-------|--------|-------|-------|-------|-----------|------|------|------|-----|------------|---------|
| Mixtral-8x7B | Pretrained | 56.91 | 84.47 | 85.29 | 64.78 | 67.03 | 35.0 | 82.43 | 70.4 | 76.16 | 69.16 |
| | SparseGPT (2:4) | 41.72 | 74.96 | 76.85 | 53.26 | 52.86 | 28.6 | 78.35 | 66.43 | 72.38 | 54.73 |
| | Wanda (2:4) | 41.55 | 74.12 | 76.61 | 53.19 | 52.26 | 27.8 | 77.04 | 63.90 | 70.48 | 59.95 |
| | NAEE (r=4) | **48.38** | 77.99 | **80.52** | 57.81 | 47.68 | 28.6 | 78.67 | 62.45 | 73.16 | 61.70 |
| | MoE-Pruner (2:4) | 47.87 | **79.00** | 79.54 | **58.86** | **62.17** | **31.8** | **79.49** | **68.23** | **74.27** | **64.58** |
| MiniCPM-8x2B | Pretrained | 42.75 | 76.22 | 77.28 | 56.49 | 52.63 | 29.0 | 77.48 | 75.81 | 66.61 | 61.58 |
| | SparseGPT (2:4) | 33.36 | 69.07 | 70.80 | 47.96 | 37.96 | 21.4 | 73.99 | 57.76 | 60.06 | 52.48 |
| | Wanda (2:4) | 33.11 | 63.34 | 66.30 | 42.31 | 27.23 | 19.6 | 69.59 | 59.57 | 55.41 | 48.50 |
| | NAEE (r=4) | 33.28 | 57.87 | 67.25 | 42.04 | 23.39 | 18.0 | 68.34 | 56.68 | 56.83 | 47.08 |
| | MoE-Pruner (2:4) | **37.71** | **71.04** | **72.54** | **51.66** | **42.42** | **24.2** | **75.08** | **70.40** | **60.62** | **56.19** |
| DeepSeek-V2-Lite | Pretrained | 46.67 | 78.28 | 79.88 | 58.65 | 54.94 | 34.2 | 80.03 | 61.37 | 71.35 | 62.81 |
| | SparseGPT (2:4) | 33.19 | 66.67 | 66.15 | 44.16 | 26.65 | 24.6 | 74.32 | 51.26 | 62.75 | 49.97 |
| | Wanda (2:4) | 31.31 | 63.97 | 65.44 | 41.85 | 30.53 | 23.2 | 72.69 | 48.01 | 61.72 | 48.75 |
| | NAEE (r=32) | 22.87 | 41.33 | 62.26 | 36.20 | 29.89 | 20.6 | 62.79 | 53.07 | 54.14 | 42.57 |
| | MoE-Pruner (2:4) | **40.02** | **71.89** | **76.61** | **50.94** | **43.85** | **27.2** | **76.22** | **55.96** | **67.64** | **56.70** |
| Qwen1.5-MoE-A2.7B | Pretrained | 41.81 | 73.32 | 79.88 | 57.98 | 61.29 | 30.0 | 80.09 | 69.31 | 68.98 | 62.58 |
| | SparseGPT (2:4) | 28.41 | **60.06** | 68.69 | 37.99 | 32.62 | 22.2 | **72.36** | 52.71 | 58.56 | 48.18 |
| | Wanda (2:4) | 28.41 | 57.45 | 63.49 | 39.12 | 28.61 | 22.6 | 71.49 | 53.79 | **62.90** | 47.58 |
| | NAEE (r=30) | 24.32 | 43.98 | 54.10 | 34.36 | 24.68 | 19.2 | 64.64 | 55.23 | 50.91 | 41.27 |
| | MoE-Pruner (2:4) | **29.10** | 52.02 | **69.14** | **42.99** | **38.14** | **25.6** | 70.57 | **55.60** | 59.12 | **49.14** |

fine-tuned Mixtral-8x7B MoE models with 50% unstructured sparsity. The fine-tuned model could achieve a 68.40 average performance on nine zero-shot tasks. The performance is very close to the pretrained Mixtral-8x7B MoE model, which demonstrates a 69.16 average performance.

Table 6: Average zero-shot performance after pruning and expert-wise knowledge distillation.

| Model | Method | ARC-c | ARC-e | Boolq | HellaSwag | MMLU | OBQA | PIQA | RTE | WinoGrande | Average |
|-------|--------|-------|-------|-------|-----------|------|------|------|-----|------------|---------|
| Mixtral -8x7B | Pretrained | 56.91 | 84.47 | 85.29 | 64.78 | 67.03 | 35.0 | 82.43 | 70.4 | 76.16 | 69.16 |
| | MoE-Pruned | 53.33 | **81.86** | **86.02** | 62.29 | 64.76 | 33.6 | 81.61 | 66.06 | 75.53 | 67.23 |
| | MoE-Distilled | **54.35** | 81.19 | 85.26 | **68.77** | 65.59 | **36.0** | 82.48 | 68.23 | **75.72** | **68.40** |

## 4.3 ABLATION STUDIES

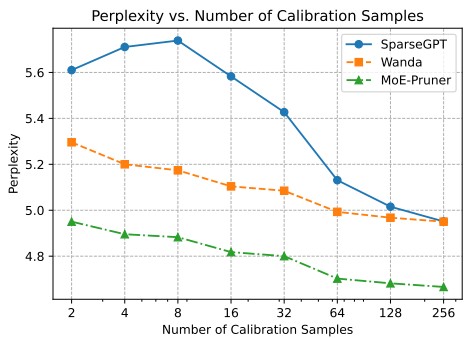

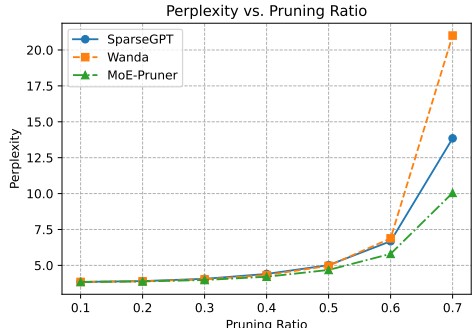

Figure 3: Perplexity with different number of calibration samples at 50% sparsity.

Figure 4: Perplexity over different pruning ratios with 128 calibration samples.

**Ablation on Different Number of Calibration Samples.** We change the number of calibration samples by selecting different sample sizes ranging from 2 to 256. Results are summarized in Figure 3. We see a clear difference in trend as the number of calibration samples changes. MoE-Pruner is much more robust than SparseGPT when there are few calibration samples and performs the same trend but better perplexity over Wanda. Notably, even with just two calibration samples, pruned networks obtained by MoE-Pruner have a perplexity of just 4.95. This may be because input norm statistics could be much easier to estimate than the full inverse Hessian of the local layer-wise reconstruction problem.

**Ablation on Different Sparsity Ratio.** We also change the pruning ratio using the same 128 calibration samples. Figure 4 shows that at lower pruning ratios, such as 10% to 40%, all pruning methods result in almost the same perplexity. When the pruning ratio increases, the Wanda pruned model perplexity changes dramatically and fails at 70%. MoE-Pruner shows better and more stable pruning results than SparseGPT and Wanda, especially at higher pruning ratios. This demonstrates that router weights preserve important information when selecting experts and provide a clear hint for pruning unimportant weights.

## 5 RELATED WORKS

**Pruning and Sparsity.** Pruning (LeCun et al., 1989; Hassibi et al., 1993; Han et al., 2015) is an important approach for compressing neural networks through eliminating weights (Han et al., 2016) or activations (Rao et al., 2021), yielding sparse networks. It can be mainly classified into two categories based on the granularity: *unstructured* and *structured* pruning.

Unstructured pruning such as magnitude pruning (Han et al., 2015; 2016) removes individual weights to introduce sparsity while preserving accuracy even at high sparsity. Existing methods either require retraining or fine-tuning the pruned models (Liu et al., 2019) or the whole iterative retraining process (Frankle & Carbin, 2019). However, in the era of LLMs, these methods fail as

retraining LLMs demands substantial computational resources. SparseGPT (Frantar & Alistarh, 2023b) and Wanda (Sun et al., 2024) propose efficient post-training pruning method that prunes LLM weights in a layer-wise manner without retraining the model.

Structured pruning eliminates weights as a group, such as channel pruning (He et al., 2017), kernel pruning (Zhong et al., 2022), attention head pruning (Wang et al., 2021), token pruning (Rao et al., 2021), and layer pruning (Elhoushi et al., 2024). Unlike unstructured pruning, it leads to more hardware-friendly, dense blocks of computation, which facilitates acceleration on modern hardware platforms. Some methods explore structured pruning based on sparsity on the structural components of LLMs, such as attention heads (Wang et al., 2021) and FFN channels (Ma et al., 2023). Muralidharan et al. (2024) uses both structured pruning and knowledge distillation to compress LLM models and shows improvement over models trained from scratch. Due to the constraint of removing regular components, structured pruning usually has low sparsity ratios and high accuracy loss. NVIDIA proposes N:M semi-structured sparsity (Mishra et al., 2021), which can preserve model performance by retraining and leverage GPU tensor core acceleration.

**Pruning for MoE Models.** Most of the works for MoE pruning focus on structured expert pruning. Chen et al. (2022) and Koishekenov et al. (2023) prune experts based on their utilization to save memory. However, this usually leads to degraded performance. Lu et al. (2024) enumerates expert combinations based on the required expert number and uses calibration data to find a set of remaining experts that has the minimum reconstruction loss. Chowdhury et al. (2024) prunes experts based on the change in the router's norm and proves that the generalization accuracy can be preserved. However, expert pruning sometimes removes experts with certain knowledge and results in the loss of model performance. Therefore, Li et al. (2024) and Zhang et al. (2024) both leverage expert merging techniques to compress the expert layer while also preserving expert knowledge. He et al. (2024) proposes a unified framework to compress MoE models. The framework consists of two perspectives: (i) expert slimming that compresses individual experts by weight pruning and quantization, and (ii) expert trimming that removes whole structured modules by layer drop and block drop.

**Efficiency for MoE and Existing Solutions.** MoE models require huge memory to host expert layers, while many experts have low utilization during inference. To address this, Gao et al. (2022) uses a tensor decomposition method to share the central tensor's parameters across experts and keep different auxiliary tensors for each expert. MoQE (Kim et al., 2023) and QMoE (Frantar & Alistarh, 2023a) both study extreme low-bit quantization for compressing MoE model size. Moreover, some works employ knowledge distillation (Fedus et al., 2022; Artetxe et al., 2021) to create either a smaller dense model or a MoE model with fewer layers. However, they also overlook the existing redundancy within MoE expert layers. Yadav et al. (2023) shows that experts can be compressed to a huge degree without any performance loss.

## 6 CONCLUSION

We propose a simple and effective pruning method for MoE models, MoE-Pruner. We prune weights with the smallest magnitudes multiplied by the corresponding input activations and router weights, on each output neuron. Our pruning method is one-shot and fast, without the need for any retraining or weight update procedures. Pruning MoE LLM with high sparsity will incur performance degradation, so we also propose a fine-tuning method that leverages the unpruned pretrained MoE model as a teacher to guide the pruned student model through expert-wise knowledge distillation. The fine-tuned MoE models could maintain 99% of the performance of the original model after the expert-wise knowledge distillation, using only a small set of training data and low GPU hours. In the future, MoE-Pruner could also be extended to structured pruning of MoE LLMs, such as channel pruning and expert pruning, for better hardware acceleration.

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

# APPENDIX

## A    MoE EXPERT ACTIVATION FREQUENCY RESULTS

### A.1    MIXTRAL-8X7B

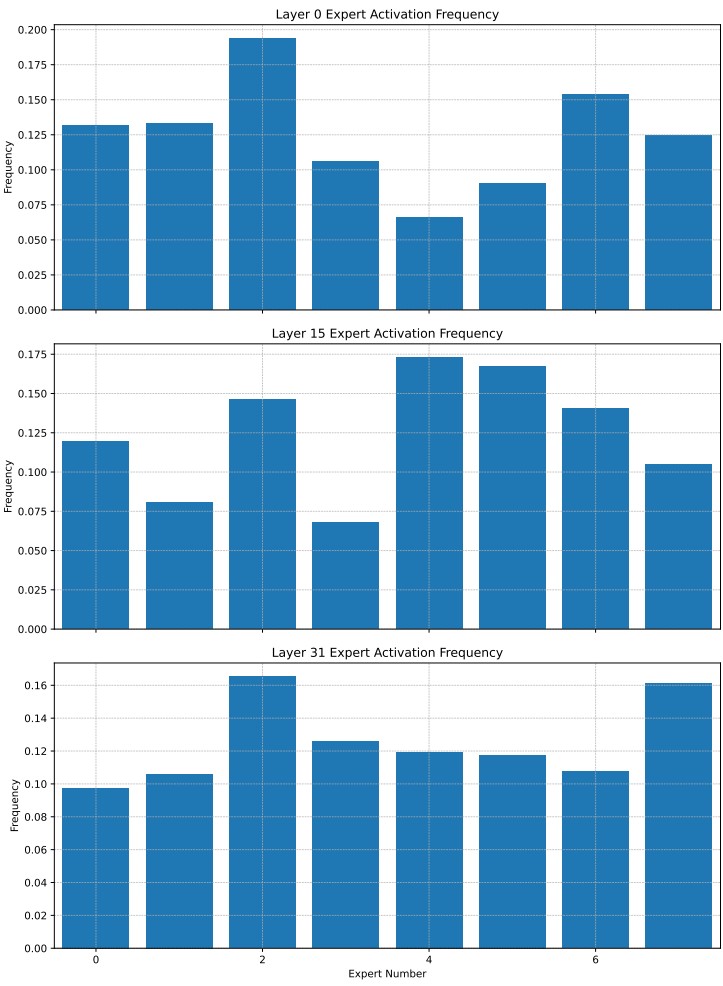

Figure 5: Mixtral-8x7B expert activation frequency on C4 datasets.

### A.2    QWEN-1.5-A2.7B

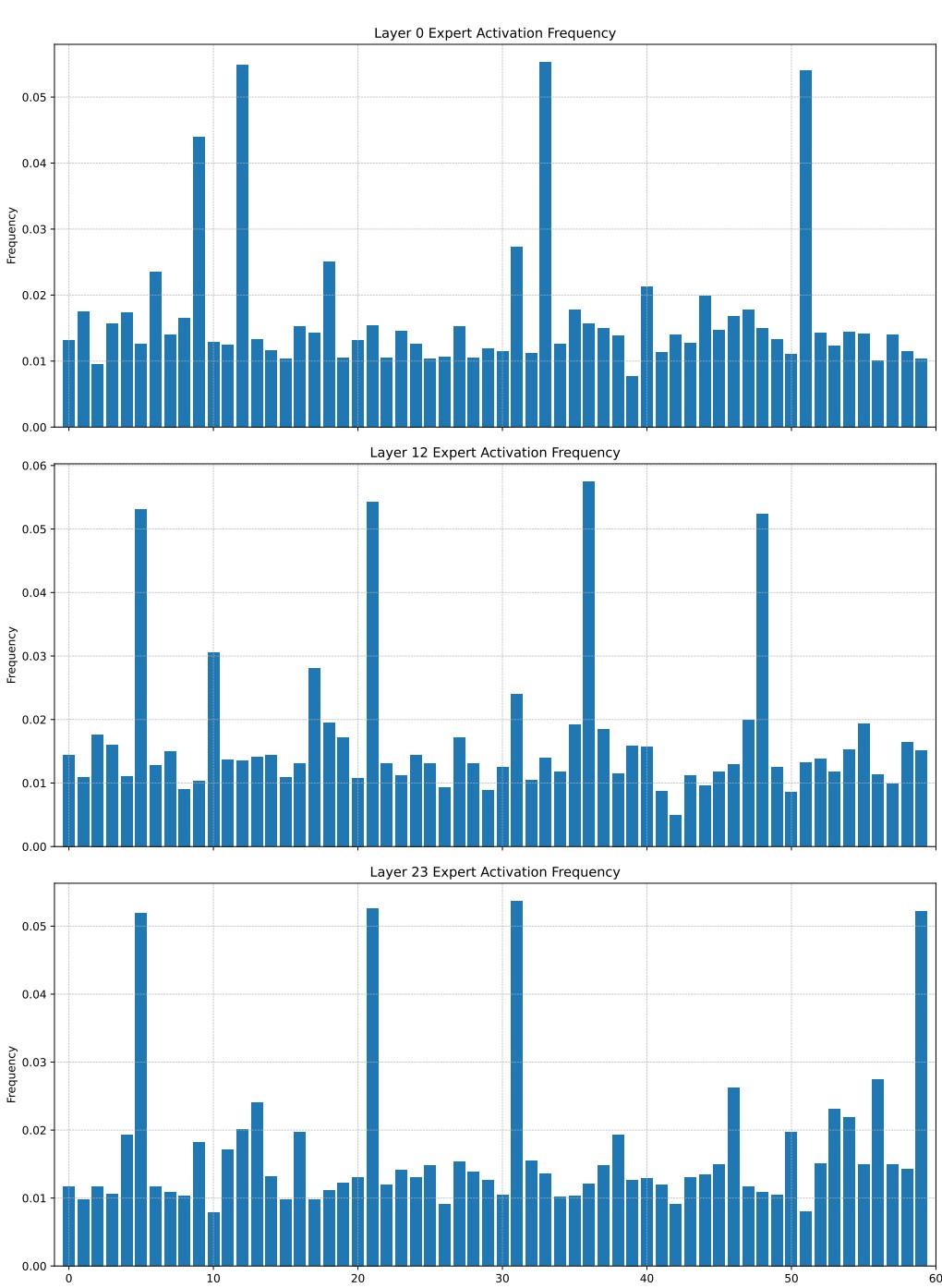

Figure 6: Qwen-1.5-A2.7B expert activation frequency on C4 datasets.

# B OPEN-SOURCE MoE MODELS

Table 7: Open-Source MoE Models List (Released after Jan. 2024).

| Name | Active Parameters | Total Parameters | # Experts | Routing Policy | Initialized Method | MMLU* |
|------|------|------|------|------|------|------|
| OLMoE | 1B | 7B | 64 | top-8 | train from scratch | 54.1 |
| MiniCPM-MoE-8x2B | 4B | 13.6B | 8 | top-2 | upcycling | 58.9 |
| Qwen1.5-MoE-A2.7B | 2.7B | 14.3B | 4(shared)+60 | 4+top-4 | upcycling | 62.5 |
| Deepseek-V2-Lite | 2.4B | 16B | 2(shared)+64 | 2+top-6 | train from scratch | 58.3 |
| Yuan2.0-M32 | 3.7B | 40B | 32 | top-2 | train from scratch | 72.2 |
| GRIN-MoE | 6.6B | 41.9B | 16 | top-2 | upcycling | 79.4 |
| Mixtral-8x7B | 12.5B | 47B | 8 | top-2 | upcycling | 70.4 |
| Jamba | 12B | 52B | 16 | top-2 | unknown | 67.4 |
| Qwen2-57B-A14B | 14B | 57.4B | 8(shared)+64 | 8+top-8 | upcycling | 76.5 |
| DBRX | 36B | 132B | 16 | top-4 | unknown | 73.7 |
| Mixtral-8x22B | 39B | 141B | 8 | top-2 | upcycling | 77.8 |
| Skywork-MoE | 22B | 146B | 16 | top-2 | upcycling | 77.4 |
| Deepseek-V2 | 21B | 236B | 2(shared)+160 | 2+top-6 | train from scratch | 78.5 |
| grok-1 | 80B | 314B | 8 | top-2 | unknown | 73.0 |
| Hunyuan-A52B | 52B | 389B | 1(shared)+16 | 1+top1 | unknown | 88.4 |
| Snowflake Arctic | 17B | 480B | 128 | top-2 | unknown | 67.3 |

*Note: This table presents a subset of open-source MoE models and is not exhaustive. The list is sorted by total parameters. MMLU scores are extracted from original papers or reports and may not reflect model real performance.

