# OpenReview forum: "MoE-Pruner: Pruning Mixture-of-Experts Large Language Model using the Hints from Its Router"
_ICLR.cc/2025/Conference — ICLR 2025 Conference Withdrawn Submission_

### Official Review · Reviewer_eqBL · 2024-11-01

**Soundness:** 3
**Presentation:** 3
**Contribution:** 2
**Rating:** 6
**Confidence:** 5

**Summary:**

This paper focuses on the pruning of sparse mixture-of-experts (MoE) models. It introduces a method called MoE-Pruner, which selectively prunes weights based on the smallest magnitudes, factoring in both the corresponding input activations and router weights for each output neuron. Experiments are conducted on the Mixtral family models to evaluate the approach's effectiveness.

**Strengths:**

(1) The method is clearly explained.

(2) The proposed method is efficient and practical for real-world deployment.

**Weaknesses:**

(1) The evaluation is restricted to the Mixtral models, raising questions about the method's scalability to other architectures like Qwen and DeepSeek.

(2) The distilled step shows only minor improvements, as seen in Table 4. What could be the underlying reason for this limited gain?

(3) This method builds incrementally on the baseline, Wanda, by incorporating gate weights within the MoE pruning framework.

**Questions:**

Some previous studies have focused on pruning entire experts, as seen in [1, 2, 3]. Could this method potentially be integrated with these approaches to enhance overall pruning effectiveness?
[1] Lu, et al. "Not All Experts are Equal: Efficient Expert Pruning and Skipping for Mixture-of-Experts Large Language Models." ACL 2024.
[2] Zhang, et al. "Diversifying the expert knowledge for task-agnostic pruning in sparse mixture-of-experts." arXiv preprint arXiv:2407.09590 (2024).
[3] Lee, Jaeseong, et al. "STUN: Structured-Then-Unstructured Pruning for Scalable MoE Pruning." arXiv preprint arXiv:2409.06211 (2024).

---

> ### Author Response · Authors · 2024-11-25
> **Response to Reviewer eqBL [1/3]**
>
> Dear reviewer eqBL,
>
> We genuinely appreciate your thorough review and constructive feedback! We have incorporated your suggestions into the revision to make the paper more compelling. Here are our responses to each question:
>
> > (1) The evaluation is restricted to the Mixtral models, raising questions about the method's scalability to other architectures like Qwen and DeepSeek.
>
> Thank you for your suggestion. To address the lack of comprehensive comparisons, we have added average zero-shot performance on 9 evaluation tasks of pruned models using SparseGPT, Wanda, and MoE-Pruner. These comparisons include **Mixtral-8x7B, MiniCPM-8x2B, DeepSeek-V2-Lite, and Qwen1.5-MoE-A2.7B** models, all pruned to **50% unstructured sparsity**:
>
> |      Model       |      Method      |   ARC-c   |   ARC-e   |   Boolq   | HellaSwag |   MMLU    |   OBQA    |   PIQA    |    RTE    | WinoGrande |  Average  |
> | :--------------: | :--------------: | :-------: | :-------: | :-------: | :-------: | :-------: | :-------: | :-------: | :-------: | :--------: | :-------: |
> |   Mixtral-8x7B   |    Pretrained    |   56.91   |   84.47   |   85.29   |   64.78   |   67.03   |   35.00   |   82.43   |   70.40   |   76.16    |   69.16   |
> |   Mixtral-8x7B   | SparseGPT (50%)  |   50.43   |   80.68   |   84.62   |   60.20   |   61.79   |   32.80   |   81.12   | **68.59** | **76.16**  |   66.27   |
> |   Mixtral-8x7B   |   Wanda (50%)    |   51.02   |   80.89   |   85.08   |   60.45   |   62.73   |   32.60   |   80.90   |   64.64   |   74.82    |   65.90   |
> |   Mixtral-8x7B   | MoE-Pruner (50%) | **53.33** | **81.86** | **86.02** | **62.29** | **64.76** | **33.60** | **81.61** |   66.06   |   75.53    | **67.23** |
> |   MiniCPM-8x2B   |    Pretrained    |   42.75   |   76.22   |   77.28   |   56.49   |   52.63   |   29.00   |   77.48   |   75.81   |   66.61    |   61.58   |
> |   MiniCPM-8x2B   | SparseGPT (50%)  |   39.25   |   73.44   | **76.36** |   53.19   |   48.35   | **28.00** |   76.22   |   64.62   | **64.96**  |   58.26   |
> |   MiniCPM-8x2B   |   Wanda (50%)    |   40.44   |   72.73   |   74.71   |   51.70   |   45.78   |   25.80   |   76.06   |   71.84   |   61.48    |   57.84   |
> |   MiniCPM-8x2B   | MoE-Pruner (50%) | **40.87** | **74.92** |   74.74   | **54.59** | **48.89** | **28.00** | **76.61** | **72.56** |   64.56    | **59.53** |
> | DeepSeek-V2-Lite |    Pretrained    |   46.67   |   78.28   |   79.88   |   58.65   |   54.94   |   34.20   |   80.03   |   61.37   |   71.35    |   62.81   |
> | DeepSeek-V2-Lite | SparseGPT (50%)  |   40.36   |   73.70   |   73.27   |   50.37   |   39.85   |   29.00   |   76.66   |   58.12   |   67.25    |   56.51   |
> | DeepSeek-V2-Lite |   Wanda (50%)    |   41.64   |   73.44   |   71.83   |   51.36   |   39.83   |   29.00   |   77.53   |   63.90   |   66.93    |   57.27   |
> | DeepSeek-V2-Lite | MoE-Pruner (50%) | **44.62** | **76.30** | **78.56** | **55.92** | **49.72** | **31.20** | **78.62** | **60.29** | **70.32**  | **60.62** |
> |   Qwen1.5-MoE    |    Pretrained    |   41.81   |   73.32   |   79.88   |   57.98   |   61.29   |   30.00   |   80.09   |   69.31   |   68.98    |   62.58   |
> |   Qwen1.5-MoE    | SparseGPT (50%)  |   34.81   |   68.90   |   76.24   |   49.86   |   51.55   |   25.20   |   77.09   |   55.96   |   67.32    |   56.33   |
> |   Qwen1.5-MoE    |   Wanda (50%)    |   33.02   |   67.30   |   75.11   |   48.26   |   50.35   |   26.80   |   75.35   |   62.09   |   65.82    |   56.01   |
> |   Qwen1.5-MoE    | MoE-Pruner (50%) | **39.68** | **72.60** | **78.44** | **54.88** | **57.63** | **30.40** | **78.73** | **72.92** | **66.93**  | **61.36** |

---

> > ### Author Response · Authors · 2024-11-25
> > **Response to Reviewer eqBL [2/3]**
> >
> > We also add average zero-shot performance on 9 evaluation tasks of pruned models using SparseGPT, Wanda, NAEE, and MoE-Pruner. These comparisons include **Mixtral-8x7B, MiniCPM-8x2B, DeepSeek-V2-Lite, and Qwen1.5-MoE-A2.7B** models, all pruned to either **2:4 structured sparsity** or **50% expert pruning**:
> >
> > |      Model       |      Method      |   ARC-c   |   ARC-e   |   Boolq   | HellaSwag |   MMLU    |   OBQA    |   PIQA    |    RTE    | WinoGrande |  Average  |
> > | :--------------: | :--------------: | :-------: | :-------: | :-------: | :-------: | :-------: | :-------: | :-------: | :-------: | :--------: | :-------: |
> > |   Mixtral-8x7B   |    Pretrained    |   56.91   |   84.47   |   85.29   |   64.78   |   67.03   |   35.00   |   82.43   |   70.40   |   76.16    |   69.16   |
> > |   Mixtral-8x7B   | SparseGPT (2:4)  |   41.72   |   74.96   |   76.85   |   53.26   |   52.86   |   28.60   |   78.35   |   66.43   |   72.38    |   54.73   |
> > |   Mixtral-8x7B   |   Wanda (2:4)    |   41.55   |   74.12   |   76.61   |   53.19   |   52.26   |   27.80   |   77.04   |   63.90   |   70.48    |   59.95   |
> > |   Mixtral-8x7B   |    NAEE (r=4)    | **48.38** |   77.99   | **80.52** |   57.81   |   47.68   |   28.60   |   78.67   |   62.45   |   73.16    |   61.70   |
> > |   Mixtral-8x7B   | MoE-Pruner (2:4) |   47.87   | **79.00** |   79.54   | **58.86** | **62.17** | **31.80** | **79.49** | **68.23** | **74.27**  | **64.58** |
> > |   MiniCPM-8x2B   |    Pretrained    |   42.75   |   76.22   |   77.28   |   56.49   |   52.63   |   29.00   |   77.48   |   75.81   |   66.61    |   61.58   |
> > |   MiniCPM-8x2B   | SparseGPT (2:4)  |   33.36   |   69.07   |   70.80   |   47.96   |   37.96   |   21.40   |   73.99   |   57.76   |   60.06    |   52.48   |
> > |   MiniCPM-8x2B   |   Wanda (2:4)    |   33.11   |   63.34   |   66.30   |   42.31   |   27.23   |   19.60   |   69.59   |   59.57   |   55.41    |   48.50   |
> > |   MiniCPM-8x2B   |    NAEE (r=4)    |   33.28   |   57.87   |   67.25   |   42.04   |   23.39   |   18.00   |   68.34   |   56.68   |   56.83    |   47.08   |
> > |   MiniCPM-8x2B   | MoE-Pruner (2:4) | **37.71** | **71.04** | **72.54** | **51.66** | **42.42** | **24.20** | **75.08** | **70.40** | **60.62**  | **56.19** |
> > | DeepSeek-V2-Lite |    Pretrained    |   46.67   |   78.28   |   79.88   |   58.65   |   54.94   |   34.20   |   80.03   |   61.37   |   71.35    |   62.81   |
> > | DeepSeek-V2-Lite | SparseGPT (2:4)  |   33.19   |   66.67   |   66.15   |   44.16   |   26.65   |   24.60   |   74.32   |   51.26   |   62.75    |   49.97   |
> > | DeepSeek-V2-Lite |   Wanda (2:4)    |   31.31   |   63.97   |   65.44   |   41.85   |   30.53   |   23.20   |   72.69   |   48.01   |   61.72    |   48.75   |
> > | DeepSeek-V2-Lite |   NAEE (r=32)*   |   22.87   |   41.33   |   62.26   |   36.20   |   29.89   |   20.60   |   62.79   |   53.07   |   54.14    |   42.57   |
> > | DeepSeek-V2-Lite | MoE-Pruner (2:4) | **40.02** | **71.89** | **76.61** | **50.94** | **43.85** | **27.20** | **76.22** | **55.96** | **67.64**  | **56.70** |
> > |   Qwen1.5-MoE    |    Pretrained    |   41.81   |   73.32   |   79.88   |   57.98   |   61.29   |   30.00   |   80.09   |   69.31   |   68.98    |   62.58   |
> > |   Qwen1.5-MoE    | SparseGPT (2:4)  |   28.41   | **60.06** |   68.69   |   37.99   |   32.62   |   22.20   | **72.36** |   52.71   |   58.56    |   48.18   |
> > |   Qwen1.5-MoE    |   Wanda (2:4)    |   28.41   |   57.45   |   63.49   |   39.12   |   28.61   |   22.60   |   71.49   |   53.79   | **62.90**  |   47.58   |
> > |   Qwen1.5-MoE    |   NAEE (r=30)*   |   24.32   |   43.98   |   54.10   |   34.36   |   24.68   |   19.20   |   64.64   |   55.23   |   50.91    |   41.27   |
> > |   Qwen1.5-MoE    | MoE-Pruner (2:4) | **29.10** |   52.02   | **69.14** | **42.99** | **38.14** | **25.60** |   70.57   | **55.60** |   59.12    | **49.14** |
> > * NAEE [1] uses an enumerate method for layer-wise expert pruning, so it only applies to the MoE model with 8 experts such as Mixtral-8x7B and MiniCPM-8x2B. For DeepSeek-V2-Lite and Qwen1.5-MoE, we use the random expert pruning.
> >
> > The updated results demonstrate that MoE-Pruner consistently outperforms SparseGPT, Wanda, and NAEE across all models and tasks. For example, in zero-shot evaluation across nine tasks, MoE-Pruner achieves:
> > - **Better task generalization**, as shown in DeepSeek-V2-Lite, where MoE-Pruner achieves an average score of 60.62 compared to SparseGPT's 56.51 and Wanda's 57.27 in 50% unstructured sparsity.
> > - **Higher average performance** than SparseGPT, Wanda, and NAEE, particularly on comprehensive benchmarks like **MMLU** (Mixtral-8x7B: 67.03 → 62.17 vs. 47.68 for NAEE in structured sparsity).
> >
> > We have also updated Table 4 and Table 5 for average zero-shot performance in our revised version.

---

> > > ### Author Response · Authors · 2024-11-25
> > > **Response to Reviewer eqBL [3/3]**
> > >
> > > > (2) The distilled step shows only minor improvements, as seen in Table 4. What could be the underlying reason for this limited gain?
> > >
> > > As shown in Section 4.2, our distillation step improves the pruned student model performance with minimal computational budget. We only use 1000 samples from the C4 dataset and the training can be done in less than one hour. We followed Wanda’s [4] idea which enforces a limited computational budget during the fine-tuning stage to be **computationally efficient while restoring pruned model performance effectively**. Considering the limited computational budget, we are able to restore the performance of pruned MoE models by a non-trivial amount.
> > >
> > > > (3) This method builds incrementally on the baseline, Wanda, by incorporating gate weights within the MoE pruning framework.
> > >
> > > Our method builds upon Wanda and extends it to MoE LLMs while maintaining the simplicity of Wanda's pruning metric. A key insight is the consideration of **router weights ($Gate$)** in MoE architectures. Consider a simple Mixture-of-Experts with two experts and each with only one weight: $y=Gate_1 \cdot w_1\cdot x + Gate_2 \cdot w_2 \cdot x$. If $\vert w_1 \vert \leq \vert w_2 \vert$, traditional pruning methods would remove $w_1$. However, in MoE architectures:
> > > - If $Gate_1 \approx 1$ and $Gate_2 = 0$, $\vert Gate_1 \cdot w_1 \cdot x \vert \gg \vert Gate_2 \cdot w_2 \cdot x \vert$, making $w_2$ the correct choice for removal.
> > >
> > > This motivating example shows that previous pruning methods for LLMs do not consider the importance of router weights which only exist in MoE architecture and may result in lower performance after pruning MoE. Therefore, we propose MoE-Pruner designed explicitly for MoE LLMs to handle such a limitation while maintaining the simplicity of Wanda's pruning metric.
> > >
> > > Thank you for your suggestion, we have added a detailed motivation and analysis of our method in the revised version, section 3.2, line 212.
> > >
> > > > Some previous studies have focused on pruning entire experts, as seen in [1, 2, 3]. Could this method potentially be integrated with these approaches to enhance overall pruning effectiveness?
> > >
> > > Yes, MoE-Pruner is compatible with expert-pruning methods and can complement them effectively:
> > > 1. **Integration with NAEE [1]:**
> > > - NAEE minimizes reconstruction loss through an enumerative expert pruning approach. MoE-Pruner could enhance NAEE by using its router weight-aware metric to guide pruning.
> > > - NAEE is limited to architectures with few experts (e.g., Mixtral-8x7B and MiniCPM-8x2B). It cannot process DeepSeek-V2-Lite and Qwen1.5-MoE with 64 or more experts.
> > > 2. **Expert Merging (e.g., [2]):**
> > > - Expert merging approaches like [2] focus on merging similar experts using a similarity matrix. MoE-Pruner’s router weights could guide the merging process by identifying underutilized experts.
> > > 3. **Integration with STUN [3]:**
> > > - STUN reduces pruning complexity for models with many experts. MoE-Pruner could be combined with STUN’s structured-then-unstructured pruning for better scalability.
> > >
> > > These integrations show that MoE-Pruner could enhance pruning efficiency and scalability.
> > >
> > > If you have other questions, we are happy to answer.
> > >
> > > References:
> > >
> > > [1]. Xudong Lu, Qi Liu, Yuhui Xu, Aojun Zhou, Siyuan Huang, Bo Zhang, Junchi Yan, Hongsheng Li. Not All Experts are Equal: Efficient Expert Pruning and Skipping for Mixture-of-Experts Large Language Models. ACL 2024.
> > >
> > > [2]. Zeliang Zhang, Xiaodong Liu, Hao Cheng, Chenliang Xu, Jianfeng Gao. Diversifying the Expert Knowledge for Task-Agnostic Pruning in Sparse Mixture-of-Experts. arXiv:2407.09590.
> > >
> > > [3]. Jaeseong Lee, Seung-won Hwang, Aurick Qiao, Daniel F Campos, Zhewei Yao, Yuxiong He. STUN: Structured-Then-Unstructured Pruning for Scalable MoE Pruning. arXiv:2409.06211.
> > >
> > > [4]. Mingjie Sun, Zhuang Liu, Anna Bair, J. Zico Kolter. A Simple and Effective Pruning Approach for Large Language Models. ICLR 2024.

---

> > > > ### Comment · Reviewer_eqBL · 2024-11-26
> > > >
> > > > Thanks a lot for the authors's response. I think this well address my concerns. The improvement on various benchmarks is significant. While it is true that the proposed method is quite similar to Wanda, the adaptive modification based on the MoE structure truly brings another performance improvement, which is surprising to me. Thus, I raise my score to 6.

---

> > > > > ### Author Response · Authors · 2024-11-26
> > > > >
> > > > > Dear reviewer eqBL,
> > > > >
> > > > > Thank you so much for your positive assessment and thoughtful response! We are delighted that our explanations and additional results addressed your concerns and demonstrated the value of our proposed method.
> > > > >
> > > > > If you have any additional suggestions or questions, we would be more than happy to address them. Thank you again for your constructive feedback and for helping us improve the quality of our paper!

---

### Official Review · Reviewer_zAPA · 2024-11-01

**Soundness:** 2
**Presentation:** 2
**Contribution:** 1
**Rating:** 3
**Confidence:** 5

**Summary:**

This work proposes MoE-Pruner, a pruning strategy for MoE based LLMs. MoE-Pruner uses the scores from the routers as signals to prune the experts and then performs an expert-wise distillation training to recover the performance. Experiments on Mixtral-8x7B and Mixtral-8x22B show that MoE-Pruner can outperform other pruning strategies for dense LLMs.

**Strengths:**

- Pruning MoE is an important and impactful research problem
- The proposed method is conceptually simple yet can achieve encouraging results

**Weaknesses:**

- **Limited technical novelty and justification** - MoE-Pruner is to be a simple extension of Wanda's formulation without extending any of its analyses to MoE. Similarly, the expert-wise knowledge distillation seems to be quite straightforward to extend from the standard knowledge distillation. Therefore, I consider the technical contribution of this work to be limited.
- **Limited evaluation** - MoE-Pruner is only evaluated on Mixtral models, it would be helpful to test its robustness by considering other models such as DeepSeek-MoE or MiniCPM-MoE.
- **Complexity analysis** - The paper does not provide a complexity analysis showing the pruning and inference speedup.
- **Poor presentation** - The paper is poorly organized. Section 2 (Preliminaries) is too short while Sections 3.1 and 3.2 are essentially a literature review, serving the same purpose. Figure 1 does not clearly explain the method as the score S does not interact with other components in the figure.

**Questions:**

- What is pruning and inference speedup of MoE-Pruner compared to other baselines?
- How did the authors extend Wanda to MoE models, is it equivalent to setting $Gate_j = 1$ in MoE-Pruner?
- The method is poorly motivated (only from L232-237). A deeper analysis (for example, Section 3, Wanda) is preferred.

---

> ### Author Response · Authors · 2024-11-25
> **Response to Reviewer zAPA [1/3]**
>
> Dear reviewer zAPA,
>
> Thank you for your thorough assessment and insightful reviews! We have carefully revised our manuscript based on your feedback. Below are our detailed responses to your comments and suggestions:
>
> > Limited technical novelty and justification - MoE-Pruner is to be a simple extension of Wanda's formulation without extending any of its analyses to MoE. Similarly, the expert-wise knowledge distillation seems to be quite straightforward to extend from the standard knowledge distillation. Therefore, I consider the technical contribution of this work to be limited.
>
> Our method extends Wanda to Mixture-of-Experts (MoE) models while maintaining the simplicity of Wanda's pruning metric. This involves incorporating **router weights** into the pruning metric, a key feature unique to MoE architectures. The motivation and theoretical formulation of MoE-Pruner explicitly account for router weights, which are critical for determining expert contributions to model outputs.
>
> Additionally, expert-wise knowledge distillation is specifically tailored for pruned MoE models. Unlike standard knowledge distillation, our method uses pretrained teacher models with identical architecture (same layers, experts, and hidden dimensions) to improve performance retention post-pruning. To the best of our knowledge, **this is the first work to introduce expert-wise knowledge distillation for MoE models**.
>
> > Limited evaluation - MoE-Pruner is only evaluated on Mixtral models, it would be helpful to test its robustness by considering other models such as DeepSeek-MoE or MiniCPM-MoE.
>
> Thank you for your suggestion. To address the lack of comprehensive comparisons, we have added average zero-shot performance on 9 evaluation tasks of pruned models using SparseGPT, Wanda, NAEE, and MoE-Pruner. These comparisons include **Mixtral-8x7B, MiniCPM-8x2B, DeepSeek-V2-Lite, and Qwen1.5-MoE-A2.7B** models, all pruned to **50% unstructured sparsity**:
>
> |      Model       |      Method      |   ARC-c   |   ARC-e   |   Boolq   | HellaSwag |   MMLU    |   OBQA    |   PIQA    |    RTE    | WinoGrande |  Average  |
> | :--------------: | :--------------: | :-------: | :-------: | :-------: | :-------: | :-------: | :-------: | :-------: | :-------: | :--------: | :-------: |
> |   Mixtral-8x7B   |    Pretrained    |   56.91   |   84.47   |   85.29   |   64.78   |   67.03   |   35.00   |   82.43   |   70.40   |   76.16    |   69.16   |
> |   Mixtral-8x7B   | SparseGPT (50%)  |   50.43   |   80.68   |   84.62   |   60.20   |   61.79   |   32.80   |   81.12   | **68.59** | **76.16**  |   66.27   |
> |   Mixtral-8x7B   |   Wanda (50%)    |   51.02   |   80.89   |   85.08   |   60.45   |   62.73   |   32.60   |   80.90   |   64.64   |   74.82    |   65.90   |
> |   Mixtral-8x7B   | MoE-Pruner (50%) | **53.33** | **81.86** | **86.02** | **62.29** | **64.76** | **33.60** | **81.61** |   66.06   |   75.53    | **67.23** |
> |   MiniCPM-8x2B   |    Pretrained    |   42.75   |   76.22   |   77.28   |   56.49   |   52.63   |   29.00   |   77.48   |   75.81   |   66.61    |   61.58   |
> |   MiniCPM-8x2B   | SparseGPT (50%)  |   39.25   |   73.44   | **76.36** |   53.19   |   48.35   | **28.00** |   76.22   |   64.62   | **64.96**  |   58.26   |
> |   MiniCPM-8x2B   |   Wanda (50%)    |   40.44   |   72.73   |   74.71   |   51.70   |   45.78   |   25.80   |   76.06   |   71.84   |   61.48    |   57.84   |
> |   MiniCPM-8x2B   | MoE-Pruner (50%) | **40.87** | **74.92** |   74.74   | **54.59** | **48.89** | **28.00** | **76.61** | **72.56** |   64.56    | **59.53** |
> | DeepSeek-V2-Lite |    Pretrained    |   46.67   |   78.28   |   79.88   |   58.65   |   54.94   |   34.20   |   80.03   |   61.37   |   71.35    |   62.81   |
> | DeepSeek-V2-Lite | SparseGPT (50%)  |   40.36   |   73.70   |   73.27   |   50.37   |   39.85   |   29.00   |   76.66   |   58.12   |   67.25    |   56.51   |
> | DeepSeek-V2-Lite |   Wanda (50%)    |   41.64   |   73.44   |   71.83   |   51.36   |   39.83   |   29.00   |   77.53   |   63.90   |   66.93    |   57.27   |
> | DeepSeek-V2-Lite | MoE-Pruner (50%) | **44.62** | **76.30** | **78.56** | **55.92** | **49.72** | **31.20** | **78.62** | **60.29** | **70.32**  | **60.62** |
> |   Qwen1.5-MoE    |    Pretrained    |   41.81   |   73.32   |   79.88   |   57.98   |   61.29   |   30.00   |   80.09   |   69.31   |   68.98    |   62.58   |
> |   Qwen1.5-MoE    | SparseGPT (50%)  |   34.81   |   68.90   |   76.24   |   49.86   |   51.55   |   25.20   |   77.09   |   55.96   |   67.32    |   56.33   |
> |   Qwen1.5-MoE    |   Wanda (50%)    |   33.02   |   67.30   |   75.11   |   48.26   |   50.35   |   26.80   |   75.35   |   62.09   |   65.82    |   56.01   |
> |   Qwen1.5-MoE    | MoE-Pruner (50%) | **39.68** | **72.60** | **78.44** | **54.88** | **57.63** | **30.40** | **78.73** | **72.92** | **66.93**  | **61.36** |

---

> > ### Author Response · Authors · 2024-11-25
> > **Response to Reviewer zAPA [2/3]**
> >
> > We also add average zero-shot performance on 9 evaluation tasks of pruned models using SparseGPT, Wanda, NAEE, and MoE-Pruner. These comparisons include **Mixtral-8x7B, MiniCPM-8x2B, DeepSeek-V2-Lite, and Qwen1.5-MoE-A2.7B** models, all pruned to either **2:4 structured sparsity** or **50% expert pruning**:
> >
> > |      Model       |      Method      |   ARC-c   |   ARC-e   |   Boolq   | HellaSwag |   MMLU    |   OBQA    |   PIQA    |    RTE    | WinoGrande |  Average  |
> > | :--------------: | :--------------: | :-------: | :-------: | :-------: | :-------: | :-------: | :-------: | :-------: | :-------: | :--------: | :-------: |
> > |   Mixtral-8x7B   |    Pretrained    |   56.91   |   84.47   |   85.29   |   64.78   |   67.03   |   35.00   |   82.43   |   70.40   |   76.16    |   69.16   |
> > |   Mixtral-8x7B   | SparseGPT (2:4)  |   41.72   |   74.96   |   76.85   |   53.26   |   52.86   |   28.60   |   78.35   |   66.43   |   72.38    |   54.73   |
> > |   Mixtral-8x7B   |   Wanda (2:4)    |   41.55   |   74.12   |   76.61   |   53.19   |   52.26   |   27.80   |   77.04   |   63.90   |   70.48    |   59.95   |
> > |   Mixtral-8x7B   |    NAEE (r=4)    | **48.38** |   77.99   | **80.52** |   57.81   |   47.68   |   28.60   |   78.67   |   62.45   |   73.16    |   61.70   |
> > |   Mixtral-8x7B   | MoE-Pruner (2:4) |   47.87   | **79.00** |   79.54   | **58.86** | **62.17** | **31.80** | **79.49** | **68.23** | **74.27**  | **64.58** |
> > |   MiniCPM-8x2B   |    Pretrained    |   42.75   |   76.22   |   77.28   |   56.49   |   52.63   |   29.00   |   77.48   |   75.81   |   66.61    |   61.58   |
> > |   MiniCPM-8x2B   | SparseGPT (2:4)  |   33.36   |   69.07   |   70.80   |   47.96   |   37.96   |   21.40   |   73.99   |   57.76   |   60.06    |   52.48   |
> > |   MiniCPM-8x2B   |   Wanda (2:4)    |   33.11   |   63.34   |   66.30   |   42.31   |   27.23   |   19.60   |   69.59   |   59.57   |   55.41    |   48.50   |
> > |   MiniCPM-8x2B   |    NAEE (r=4)    |   33.28   |   57.87   |   67.25   |   42.04   |   23.39   |   18.00   |   68.34   |   56.68   |   56.83    |   47.08   |
> > |   MiniCPM-8x2B   | MoE-Pruner (2:4) | **37.71** | **71.04** | **72.54** | **51.66** | **42.42** | **24.20** | **75.08** | **70.40** | **60.62**  | **56.19** |
> > | DeepSeek-V2-Lite |    Pretrained    |   46.67   |   78.28   |   79.88   |   58.65   |   54.94   |   34.20   |   80.03   |   61.37   |   71.35    |   62.81   |
> > | DeepSeek-V2-Lite | SparseGPT (2:4)  |   33.19   |   66.67   |   66.15   |   44.16   |   26.65   |   24.60   |   74.32   |   51.26   |   62.75    |   49.97   |
> > | DeepSeek-V2-Lite |   Wanda (2:4)    |   31.31   |   63.97   |   65.44   |   41.85   |   30.53   |   23.20   |   72.69   |   48.01   |   61.72    |   48.75   |
> > | DeepSeek-V2-Lite |   NAEE (r=32)*   |   22.87   |   41.33   |   62.26   |   36.20   |   29.89   |   20.60   |   62.79   |   53.07   |   54.14    |   42.57   |
> > | DeepSeek-V2-Lite | MoE-Pruner (2:4) | **40.02** | **71.89** | **76.61** | **50.94** | **43.85** | **27.20** | **76.22** | **55.96** | **67.64**  | **56.70** |
> > |   Qwen1.5-MoE    |    Pretrained    |   41.81   |   73.32   |   79.88   |   57.98   |   61.29   |   30.00   |   80.09   |   69.31   |   68.98    |   62.58   |
> > |   Qwen1.5-MoE    | SparseGPT (2:4)  |   28.41   | **60.06** |   68.69   |   37.99   |   32.62   |   22.20   | **72.36** |   52.71   |   58.56    |   48.18   |
> > |   Qwen1.5-MoE    |   Wanda (2:4)    |   28.41   |   57.45   |   63.49   |   39.12   |   28.61   |   22.60   |   71.49   |   53.79   | **62.90**  |   47.58   |
> > |   Qwen1.5-MoE    |   NAEE (r=30)*   |   24.32   |   43.98   |   54.10   |   34.36   |   24.68   |   19.20   |   64.64   |   55.23   |   50.91    |   41.27   |
> > |   Qwen1.5-MoE    | MoE-Pruner (2:4) | **29.10** |   52.02   | **69.14** | **42.99** | **38.14** | **25.60** |   70.57   | **55.60** |   59.12    | **49.14** |
> > * NAEE [1] uses an enumerate method for layer-wise expert pruning, so it only applies to the MoE model with 8 experts such as Mixtral-8x7B and MiniCPM-8x2B. For DeepSeek-V2-Lite and Qwen1.5-MoE, we use the random expert pruning.
> >
> > The updated results demonstrate that MoE-Pruner consistently outperforms SparseGPT, Wanda, and NAEE across all models and tasks. For example, in zero-shot evaluation across nine tasks, MoE-Pruner achieves:
> > - **Better task generalization**, as shown in DeepSeek-V2-Lite, where MoE-Pruner achieves an average score of 60.62 compared to SparseGPT's 56.51 and Wanda's 57.27 in 50% unstructured sparsity.
> > - **Higher average performance** than SparseGPT, Wanda, and NAEE, particularly on comprehensive benchmarks like **MMLU** (Mixtral-8x7B: 67.03 → 62.17 vs. 47.68 for NAEE in structured sparsity).
> >
> > We have also updated Table 4 and Table 5 for average zero-shot performance in our revised version.

---

> > > ### Author Response · Authors · 2024-11-25
> > > **Response to Reviewer zAPA [3/3]**
> > >
> > > > Complexity analysis - The paper does not provide a complexity analysis showing the pruning and inference speedup.
> > >
> > > We conduct an **efficiency evaluation experiment** on memory reduction and inference speedup using the A100 GPU. To provide a comprehensive comparison, we also evaluated the NAEE [1] expert pruning method. Comparison results are shown in the table below. Our proposed MoE-Pruner method at the structured 2:4 sparsity pattern outperforms NAEE in terms of both average performance and inference speedup, while incurring only a small memory overhead for storing sparse tensor indices. The original average performance and memory of Mixtral-8x7B are 69.16 and 87.49GB, respectively. Speedup is measured according to the original Mixtral 8x7B baseline. The following table has been added to the Table 3 efficiency evaluation experiments in our revised version:
> > >
> > > | Model | Method | Sparsity | Average | Memory(GB) | Speedup |
> > > | ----------- | ----------- | ----------- | ----------- | ----------- | ----------- |
> > > | Mixtral-8x7B | NAEE[1] | r=4 | 61.69 | 45.49 | 1.06× |
> > > | Mixtral-8x7B | MoE-Pruner | 2:4 | 64.58 | 50.74 | 1.14× |
> > >
> > > > Poor presentation - The paper is poorly organized. Section 2 (Preliminaries) is too short while Sections 3.1 and 3.2 are essentially a literature review, serving the same purpose. Figure 1 does not clearly explain the method as the score S does not interact with other components in the figure.
> > >
> > > We have restructured the manuscript as follows:
> > > - **Section 2 (Preliminaries)**: Expanded to include details on MoE initialization methods, including sparse upcycling initialization [2], and its relevance to MoE pruning.
> > > - **Section 3 (Methodology)**: Streamlined to focus on MoE-Pruner’s formulation and motivation. We included a detailed motivating example that demonstrates how router weights influence pruning decisions in MoE architectures.
> > > - **Figure 1**: Removed to avoid redundancy. Instead, we emphasize how the pruning metric ($S_{ij}$) directly interacts with other components.
> > >
> > > > What is pruning and inference speedup of MoE-Pruner compared to other baselines?
> > >
> > > Please refer to our response to weakness 3 for the efficiency evaluation experiment about memory reduction and inference speedup.
> > >
> > > > How did the authors extend Wanda to MoE models, is it equivalent to setting Gate_j=1 in MoE-Pruner?
> > >
> > > Yes, if we set Gate value for each expert to be 1 in MoE-Pruner, then our method becomes the same as Wanda.
> > >
> > > > The method is poorly motivated (only from L232-237). A deeper analysis (for example, Section 3, Wanda) is preferred.
> > >
> > > Our method builds upon Wanda and extends it to MoE LLMs while maintaining the simplicity of Wanda's pruning metric. A key insight is the consideration of **router weights** ($Gate$) in MoE architectures. Consider a simple Mixture-of-Experts with two experts and each with only one weight: $y=Gate_1 \cdot w_1 \cdot x + Gate_2 \cdot w_2 \cdot x$. If $\vert w_1 \vert \leq \vert w_2 \vert$, traditional pruning methods would remove $w_1$. However, in MoE architectures:
> > > - If $Gate_1 \approx 1$ and $Gate_2 = 0$, $ \vert Gate_1 \cdot w_1 \cdot x \vert \gg \vert Gate_2 \cdot w_2 \cdot x \vert$, making $w_2$ the correct choice for removal.
> > >
> > > This motivating example shows that previous pruning methods for LLMs do not consider the importance of router weights which only exist in MoE architecture and may result in lower performance after pruning MoE. Therefore, we propose MoE-Pruner designed explicitly for MoE LLMs to handle such a limitation while maintaining the simplicity of Wanda's pruning metric.
> > >
> > > Thank you for your suggestion, we have added a detailed motivation and analysis of our method in the revised version, section 3.2, line 212.
> > >
> > > If you have other questions, we are happy to answer.
> > >
> > > References
> > >
> > > [1]. Xudong Lu, Qi Liu, Yuhui Xu, Aojun Zhou, Siyuan Huang, Bo Zhang, Junchi Yan, Hongsheng Li. Not All Experts are Equal: Efficient Expert Pruning and Skipping for Mixture-of-Experts Large Language Models. ACL 2024.
> > >
> > > [2] Aran Komatsuzaki, Joan Puigcerver, James Lee-Thorp, Carlos Riquelme Ruiz, Basil Mustafa, Joshua Ainslie, Yi Tay, Mostafa Dehghani, Neil Houlsby. Sparse Upcycling: Training Mixture-of-Experts from Dense Checkpoints. ICLR 2023.

---

> ### Author Response · Authors · 2024-12-03
>
> Dear Reviewer zAPA,
>
> Thank you once again for your time and thoughtful feedback on our submission. We hope that our responses to your comments have addressed your concerns.
>
> As the rebuttal phase is nearing its conclusion, we wanted to kindly follow up to check if there are any additional questions that you would like us to clarify. Thank you again for your time and effort in reviewing our paper!

---

### Official Review · Reviewer_qxdo · 2024-11-04

**Soundness:** 2
**Presentation:** 2
**Contribution:** 2
**Rating:** 5
**Confidence:** 4

**Summary:**

In this paper, the authors propose a simple and effective pruning method for MoE models. Specifically, (1) the authors prune model weights based on the sensitivity criterion derived from the MoE gate outputs in each transformer block; (2) they further use a distillation method to recover task performance of the pruned model. Experimental results show that on zero-shot and language modeling datasets, the proposed method outperforms existing weight pruning methods in terms of algorithm performance.

**Strengths:**

1. The authors propose a simple yet effective MoE pruning method that can be easily applied to various existing MoE models, achieving notable performance improvements.
  2. The authors provide a detailed ablation study of each component and hyperparameter, clearly demonstrating the role of each component and the sensitivity to hyperparameters.

**Weaknesses:**

1. Equations lack detailed explanation.

    - For equations 7, 8, and 9, what do i and j stand for? Please describe in detail which dimension in each tensor corresponds to i and j. By the way, I also notice that in equation 9, the authors use S instead of S_{i,j} to represent the sensitivity metric. Is this a typo?

    - For equation 10, is cross entropy the global next token prediction loss or the local cross entropy loss for the MoE gate?

2. Algorithm 1 needs detailed explanation. In line 251, the authors mentioned that "Algorithm 1 presents the unstructured sparsity version of our MoE-Pruner algorithm". I don’t understand where "unstructured" is reflected. Is it that the weights of the experts are unstructured? Does the weight WWW represent all experts or just one expert? Also, which linear layer in SwiGLU does the weight WWW correspond to?

3. The experimental comparisons are not comprehensive. In the Introduction, the authors mention various issues with expert merging and expert pruning. However, in the experimental section, they do not compare their method with any expert merging or pruning methods [1,2,3]. The authors need to provide some comparative data and analysis to demonstrate whether their proposed method is truly SOTA.

[1] Liu et al., Efficient expert pruning for sparse mixture-of-experts language models: Enhancing performance and reducing inference costs

[2] Lu et al., Not All Experts are Equal: Efficient Expert Pruning and Skipping for Mixture-of-Experts Large Language Models

[3] Muzio et al., SEER-MoE: Sparse Expert Efficiency through Regularization for Mixture-of-Experts

**Questions:**

1. In line 154, the authors mentioned that upcycling initialization will lead to higher expert similarity in MoE models. Since the similarity between experts is relatively high, pruning experts should theoretically not result in a significant performance drop. Why do the authors reach the opposite conclusion?
  2. In line 158, the authors mentioned that train from stratch will show imbalanced expert activation frequency, indicating that least-used expert pruning could help compress model size and not bring performance degradation. However, I believe this may be task-dependent; these seemingly least-used experts could be very useful for specific tasks such as math or coding. Did the authors conduct any related validation?

---

> ### Author Response · Authors · 2024-11-25
> **Response to Reviewer qxdo [1/3]**
>
> Dear reviewer qxdo,
>
> We really appreciate your thorough and constructive feedback. We have taken your suggestions to heart and made changes to the best of our abilities. We respond to your questions and comments:
>
>
> > 1.1 For equations 7, 8, and 9, what do i and j stand for? Please describe in detail which dimension in each tensor corresponds to i and j. By the way, I also notice that in equation 9, the authors use S instead of S_{i,j} to represent the sensitivity metric. Is this a typo?
>
> For equations 7, 8, and 9, the $i$ and $j$ stands for **output feature** and **input feature dimension**, respectively. Thank you for pointing out the use of $S$ and $S_{ij}$. We have revised all equations of pruning metric to $S_{ij}$ in the revised version for better clarity.
>
> > 1.2 For equation 10, is cross entropy the global next token prediction loss or the local cross entropy loss for the MoE gate?
>
> For equation 10, the first cross entropy loss is the global token-level cross-entropy loss for next-token prediction tasks, while the second term refers to the expert-wise knowledge distillation loss, computed as the summation of each expert’s local MSE loss.
>
> > 2. Algorithm 1 needs detailed explanation. In line 251, the authors mentioned that "Algorithm 1 presents the unstructured sparsity version of our MoE-Pruner algorithm". I don’t understand where "unstructured" is reflected. Is it that the weights of the experts are unstructured? Does the weight WWW represent all experts or just one expert? Also, which linear layer in SwiGLU does the weight WWW correspond to?
>
> - For algorithm 1, the unstructured sparsity refers to the fact that the locations of pruned weights are not constrained. In our experiment, we target each linear layer **output neuron** $W_i$ to be 50% sparse, which ensures each weight tensor $W_{ij}$ is also 50% sparse.
> For 2:4 semi structured sparsity, we prune 2 weight elements out of 4 consecutive weight elements in the output neuron $W_i$, which adds more constraints.
> As you can see in Algorithm 1 line 8, we set each output neuron $W_i$ that has $d_{col}$ weight elements to be p% sparse. The pruning results are pruned expert weights.
>
> - Sorry we do not have weight WWW in Algorithm 1. If you mean line 9: $ W \gets M \odot W $. We apply the pruning mask $M$ (with 0 for pruning and 1 remaining) to the weights $W$ and get the final pruned weights. Yes, we target every expert weight.
>
> - For all MoE models which uses SwiGLU architecture, we prune 50% for all three weights, including gate_proj, up_proj, and down_proj.

---

> > ### Author Response · Authors · 2024-11-25
> > **Response to Reviewer qxdo [2/3]**
> >
> > > 3. The experimental comparisons are not comprehensive. In the Introduction, the authors mention various issues with expert merging and expert pruning. However, in the experimental section, they do not compare their method with any expert merging or pruning methods [1,2,3]. The authors need to provide some comparative data and analysis to demonstrate whether their proposed method is truly SOTA.
> >
> > Certainly. To address the lack of comprehensive comparisons, we have added average zero-shot performance on 9 evaluation tasks of pruned models using SparseGPT, Wanda, NAEE [2], and MoE-Pruner. These comparisons include **Mixtral-8x7B, MiniCPM-8x2B, DeepSeek-V2-Lite, and Qwen1.5-MoE-A2.7B** models, all pruned to either **2:4 structured sparsity** or **50% expert pruning**:
> >
> > |      Model       |      Method      |   ARC-c   |   ARC-e   |   Boolq   | HellaSwag |   MMLU    |   OBQA    |   PIQA    |    RTE    | WinoGrande |  Average  |
> > | :--------------: | :--------------: | :-------: | :-------: | :-------: | :-------: | :-------: | :-------: | :-------: | :-------: | :--------: | :-------: |
> > |   Mixtral-8x7B   |    Pretrained    |   56.91   |   84.47   |   85.29   |   64.78   |   67.03   |   35.00   |   82.43   |   70.40   |   76.16    |   69.16   |
> > |   Mixtral-8x7B   | SparseGPT (2:4)  |   41.72   |   74.96   |   76.85   |   53.26   |   52.86   |   28.60   |   78.35   |   66.43   |   72.38    |   54.73   |
> > |   Mixtral-8x7B   |   Wanda (2:4)    |   41.55   |   74.12   |   76.61   |   53.19   |   52.26   |   27.80   |   77.04   |   63.90   |   70.48    |   59.95   |
> > |   Mixtral-8x7B   |    NAEE (r=4)    | **48.38** |   77.99   | **80.52** |   57.81   |   47.68   |   28.60   |   78.67   |   62.45   |   73.16    |   61.70   |
> > |   Mixtral-8x7B   | MoE-Pruner (2:4) |   47.87   | **79.00** |   79.54   | **58.86** | **62.17** | **31.80** | **79.49** | **68.23** | **74.27**  | **64.58** |
> > |   MiniCPM-8x2B   |    Pretrained    |   42.75   |   76.22   |   77.28   |   56.49   |   52.63   |   29.00   |   77.48   |   75.81   |   66.61    |   61.58   |
> > |   MiniCPM-8x2B   | SparseGPT (2:4)  |   33.36   |   69.07   |   70.80   |   47.96   |   37.96   |   21.40   |   73.99   |   57.76   |   60.06    |   52.48   |
> > |   MiniCPM-8x2B   |   Wanda (2:4)    |   33.11   |   63.34   |   66.30   |   42.31   |   27.23   |   19.60   |   69.59   |   59.57   |   55.41    |   48.50   |
> > |   MiniCPM-8x2B   |    NAEE (r=4)    |   33.28   |   57.87   |   67.25   |   42.04   |   23.39   |   18.00   |   68.34   |   56.68   |   56.83    |   47.08   |
> > |   MiniCPM-8x2B   | MoE-Pruner (2:4) | **37.71** | **71.04** | **72.54** | **51.66** | **42.42** | **24.20** | **75.08** | **70.40** | **60.62**  | **56.19** |
> > | DeepSeek-V2-Lite |    Pretrained    |   46.67   |   78.28   |   79.88   |   58.65   |   54.94   |   34.20   |   80.03   |   61.37   |   71.35    |   62.81   |
> > | DeepSeek-V2-Lite | SparseGPT (2:4)  |   33.19   |   66.67   |   66.15   |   44.16   |   26.65   |   24.60   |   74.32   |   51.26   |   62.75    |   49.97   |
> > | DeepSeek-V2-Lite |   Wanda (2:4)    |   31.31   |   63.97   |   65.44   |   41.85   |   30.53   |   23.20   |   72.69   |   48.01   |   61.72    |   48.75   |
> > | DeepSeek-V2-Lite |   NAEE (r=32)*   |   22.87   |   41.33   |   62.26   |   36.20   |   29.89   |   20.60   |   62.79   |   53.07   |   54.14    |   42.57   |
> > | DeepSeek-V2-Lite | MoE-Pruner (2:4) | **40.02** | **71.89** | **76.61** | **50.94** | **43.85** | **27.20** | **76.22** | **55.96** | **67.64**  | **56.70** |
> > |   Qwen1.5-MoE    |    Pretrained    |   41.81   |   73.32   |   79.88   |   57.98   |   61.29   |   30.00   |   80.09   |   69.31   |   68.98    |   62.58   |
> > |   Qwen1.5-MoE    | SparseGPT (2:4)  |   28.41   | **60.06** |   68.69   |   37.99   |   32.62   |   22.20   | **72.36** |   52.71   |   58.56    |   48.18   |
> > |   Qwen1.5-MoE    |   Wanda (2:4)    |   28.41   |   57.45   |   63.49   |   39.12   |   28.61   |   22.60   |   71.49   |   53.79   | **62.90**  |   47.58   |
> > |   Qwen1.5-MoE    |   NAEE (r=30)*   |   24.32   |   43.98   |   54.10   |   34.36   |   24.68   |   19.20   |   64.64   |   55.23   |   50.91    |   41.27   |
> > |   Qwen1.5-MoE    | MoE-Pruner (2:4) | **29.10** |   52.02   | **69.14** | **42.99** | **38.14** | **25.60** |   70.57   | **55.60** |   59.12    | **49.14** |
> > * NAEE [2] uses an enumerate method for layer-wise expert pruning, so it only applies to the MoE model with 8 experts such as Mixtral-8x7B and MiniCPM-8x2B. For DeepSeek-V2-Lite and Qwen1.5-MoE, we use the random expert pruning.
> >
> > [3] does not provide code or model for comparison. [1] uses a different framework and only releases weights for Mixtral-8x7B-Instruct and we only share two same evaluation datasets: Boolq and RTE. However, the reported results in [1] for Mixtral-8x7B-Instruct on Boolq is 73.2. While NAEE [2] and our tested results both report 88.5. Thus we may not be able to compare our results with [1].

---

> > > ### Author Response · Authors · 2024-11-25
> > > **Response to Reviewer qxdo [3/3]**
> > >
> > > (Continued response to weakness 3)
> > >
> > > Compared with the expert pruning method NAEE [2], our method shows much higher average performance across all four MoE models at 50% structured sparsity. Especially if you look at the MMLU performance, which is the most comprehensive benchmark we used, expert pruning results in a very large performance degradation (Mixtral-8x7B 67.03 to 47.68, MiniCPM-8x2B 56.23 to 23.39), while our method shows a moderate loss (Mixtral-8x7B 67.03 to 62.17, MiniCPM-8x2B 56.23 to 42.42).
> > >
> > > > 1. In line 154, the authors mentioned that upcycling initialization will lead to higher expert similarity in MoE models. Since the similarity between experts is relatively high, pruning experts should theoretically not result in a significant performance drop. Why do the authors reach the opposite conclusion?
> > >
> > > This is a very good and interesting question. Sparse upcycled MoE models start from the same expert weight and have a higher similarity between experts. Pruning experts seem to have a lower performance drop. However, as demonstrated in our experiments (Table above), expert pruning results in severe degradation (e.g., Mixtral-8x7B MMLU: 67.03 → 47.68), while our 2:4 structured pruning exhibits a much smaller loss (67.03 → 62.17).
> > >
> > > The reason is that sparse upcycled MoE models usually use **a smaller number of experts (e.g. 8 experts)** and have a more **uniform expert activation frequency**. If we consider a top-2 activation method with 8 experts, then each expert will process 25% of the input tokens. Pruning half of the expert will result in severe performance drop. Our 2:4 structured pruning tries to preserve each expert performance while maintaining a 50% structured sparsity. Experimental results show that our methods achieve better performance on sparse upcycled MoE models compared with expert pruning method.
> > >
> > > > 2. In line 158, the authors mentioned that train from scratch will show imbalanced expert activation frequency, indicating that least-used expert pruning could help compress model size and not bring performance degradation. However, I believe this may be task-dependent; these seemingly least-used experts could be very useful for specific tasks such as math or coding. Did the authors conduct any related validation?
> > >
> > > Thank you for your insightful suggestion. For a model trained from scratch, we use DeepSeek-V2-Lite and test its performance on a representative math datasets, GSM8K, with NAEE [2] and MoE-Pruner:
> > > |      Model       |            Method            | Sparsity | GSM8K(5-shot) | Average Performance |
> > > | :--------------: | :--------------------------: | :------: | :-----------: | :-----------------: |
> > > | DeepSeek-V2-Lite |          Pretrained          |    -     |     38.51     |        62.81        |
> > > | DeepSeek-V2-Lite | NAEE (Random expert pruning) |   r=32   |     0.99      |        42.57        |
> > > | DeepSeek-V2-Lite |          MoE-Pruner          |   2:4    |     9.17      |        56.70        |
> > >
> > > Results demonstrate that expert pruning (NAEE) causes a drastic performance drop (GSM8K: 38.51 → 0.99), while MoE-Pruner retains better performance (38.51 → 9.17). We use random expert pruning as NAEE [2] uses an enumerate method, so it only applies to the MoE model with a smaller number of experts, e.g. 8, but not 64. This large performance drop is also proven in the NAEE [2] paper Table 3, suggesting that expert pruning may be better for task-dependent setting or followup fine-tuning scenarios. We will incorporate this finding into our draft.
> > >
> > > If you have other questions, we are happy to answer.
> > >
> > > References:
> > >
> > > [1]. Enshu Liu, Junyi Zhu, Zinan Lin, Xuefei Ning, Matthew B. Blaschko, Shengen Yan, Guohao Dai, Huazhong Yang, Yu Wang. Efficient Expert Pruning for Sparse Mixture-of-Experts Language Models: Enhancing Performance and Reducing Inference Costs. arXiv:2407.00945.
> > >
> > > [2]. Xudong Lu, Qi Liu, Yuhui Xu, Aojun Zhou, Siyuan Huang, Bo Zhang, Junchi Yan, Hongsheng Li. Not All Experts are Equal: Efficient Expert Pruning and Skipping for Mixture-of-Experts Large Language Models. ACL 2024.
> > >
> > > [3]. Alexandre Muzio, Alex Sun, Churan He. SEER-MoE: Sparse Expert Efficiency through Regularization for Mixture-of-Experts. arXiv:2404.05089.

---

> > > > ### Comment · Reviewer_qxdo · 2024-11-29
> > > >
> > > > Thanks for the rebuttal and the additional experiments. I have a clearer understanding of the revised paper. However, I still have some concerns:
> > > >
> > > > 1. **Clarity Issues:** There are still clarity problems in the paper that need to be addressed. For example, in Equation 9, the authors may use **gate_k** instead of **gate** to differentiate between different experts; otherwise, they do not correspond to gate_1 and gate_2 mentioned in line 218. Additionally, what do *i* and *j* represent in Equations 7 and 8? Do the *i* in Equations 7, 8, and 9 overlap with the *i* in Equation 1? Similar issues should be carefully reviewed and resolved.
> > > >
> > > > 2. **Method Design:** The current method does capture an interesting point, but its overall method design remains relatively simple and not that effective (based on the experiments). For instance, there are many other intriguing approaches for 2:4 sparsity, such as channel permutation [1]. I suggest that the authors consider combining the **gate** value with other approaches, **modeling the overall method as an optimization problem**, which might result in a more solid method design and evaluation results. Furthermore, the distillation approach does not demonstrate significant differences compared to existing distillation methods. Could the lambda in Equation 10 be designed to depend on the gate scores?
> > > >
> > > > 3. **Experimental Results:** Based on the current experimental results, the algorithm’s performance still falls significantly short of the FP16 model, particularly on challenging benchmarks like MMLU. The pruned Qwen1.5-MoE model exhibits an accuracy loss of over 20%. I suggest that the authors further refine the proposed methods to better align with the characteristics of MoE models, aiming to improve the performance.
> > > >
> > > > [1] Pool, Jeff, and Chong Yu. "Channel permutations for n: m sparsity." Advances in neural information processing systems 34 (2021): 13316-13327.

---

> > > > > ### Author Response · Authors · 2024-12-03
> > > > > **Response to Reviewer qxdo [1/2]**
> > > > >
> > > > > Thank you for looking into our rebuttal and providing insightful suggestions to improve our paper quality. Here are our response to your followup questions:
> > > > >
> > > > > > Clarity Issues: There are still clarity problems in the paper that need to be addressed. For example, in Equation 9, the authors may use gate_k instead of gate to differentiate between different experts; otherwise, they do not correspond to gate_1 and gate_2 mentioned in line 218. Additionally, what do i and j represent in Equations 7 and 8? Do the i in Equations 7, 8, and 9 overlap with the i in Equation 1? Similar issues should be carefully reviewed and resolved.
> > > > >
> > > > > Thank you for pointing out the clarity issues. Yes, we should use $Gate_k$ to represent corresponding router weights to the k-th expert.
> > > > > The $i$ and $j$ in Equation 7 and 8 stand for output feature and input feature dimension, which is the same as Equation 9.
> > > > > To resolve the clarity issue with $i$, we should change those $i$ in Equation 1, 3, 4, and 5 to $k$ to represent the corresponding expert index:
> > > > >
> > > > > Equation 1: $y=\sum_{k=0}^{n-1} Gate(x)_k \cdot E_k(x)$
> > > > >
> > > > > Equation 3: $y=\sum_{k=0}^{n-1} \mathrm{Softmax}(\mathrm{Top2}(x\cdot W_g))_k \cdot \mathrm{SwiGLU}_k(x)$
> > > > >
> > > > > Equation 4: $s = \frac{\sigma}{\mu} = \frac{\sqrt{\frac{1}{n}\sum_{k=0}^{n-1}(f_k-\mu)^2}}{\mu}$ $\mu = \frac{1}{n}\sum_{k=0}^{n-1}f_k$
> > > > >
> > > > > Equation 5: $f_k = \sum_{x \in \mathcal{B}} {1} \{\text{argmax}\: p(x) = k\}$
> > > > >
> > > > > We will incorporate these changes into our draft.
> > > > >
> > > > >
> > > > > > Method Design: The current method does capture an interesting point, but its overall method design remains relatively simple and not that effective (based on the experiments). For instance, there are many other intriguing approaches for 2:4 sparsity, such as channel permutation [1]. I suggest that the authors consider combining the gate value with other approaches, modeling the overall method as an optimization problem, which might result in a more solid method design and evaluation results. Furthermore, the distillation approach does not demonstrate significant differences compared to existing distillation methods. Could the lambda in Equation 10 be designed to depend on the gate scores?
> > > > >
> > > > > The channel permutation [1] is a method to maximize the accuracy of N:M sparse networks. It could complement our 2:4 pruning strategy and offers performance improvement of 2:4 sparse networks. This technique is proposed by [1] and revisited for LLM pruning recently in [2]. Since SparseGPT and Wanda in 2:4 sparse settings did not include channel permutation technique, we also implement our method for 2:4 structured sparsity without channel permutation to ensure fair comparison. To achieve better model performance after pruning, we will add channel permutation technique for 2:4 sparse settings in the future.
> > > > >
> > > > > For Equation 10, setting the $\lambda$ value to be dependent on the gate scores is interesting. In our experiment, we sum over all MSE losses between pretrained teacher experts and pruned student experts. We find that setting the $\lambda$ to be the cross entropy loss over the expert-wise knowledge distillation loss could guarantee convergence and better performance after training. We could actually assign  different $\lambda$ values for different experts depending on the router weights or gate scores. This could help expert diversification and potentially improve model performance after distillation. We will conduct experiments to evaluate the effectiveness of dynamically assigning $\lambda$ based on gate scores and include these results in future work.

---

> > > > > ### Author Response · Authors · 2024-12-03
> > > > > **Response to Reviewer qxdo [2/2]**
> > > > >
> > > > > > Experimental Results: Based on the current experimental results, the algorithm’s performance still falls significantly short of the FP16 model, particularly on challenging benchmarks like MMLU. The pruned Qwen1.5-MoE model exhibits an accuracy loss of over 20%. I suggest that the authors further refine the proposed methods to better align with the characteristics of MoE models, aiming to improve the performance.
> > > > >
> > > > > As shown in the experimental results for Qwen1.5-MoE-A2.7B model with 2:4 structured sparsity or 50% expert pruning, our method MoE-Pruner outperforms all previous methods, such as SparseGPT, Wanda, and NAEE. The performance drop in MMLU and other benchmarks shows that this is a model-related scenario. As show in our paper Table 1 caption:
> > > > > > The only exception is Qwen-1.5-A2.7B, which is initialized with upcycling. But according to the report [3], its expert parameters are shuffled along the intermediate dimension to guarantee that each fine-grained expert exhibits unique characteristics.
> > > > >
> > > > > We reinvestigated this Qwen1.5-MoE-A2.7B model (4 shared experts and 60 routed experts) and found that pruning the shared expert results in severe performance drop, while preserving the shared expert will improve the pruned model performance by a large margin. The shared expert contributes to 6% of model parameters, which means that for 50% sparsity, if we keep the shared expert, our method only loses 3% sparsity but improves the model performance by more than 10% and especially on MMLU by more than 18%. We list the average zero-shot performance of pruned models while keeping the shared expert:
> > > > >
> > > > > |       Model       |   Method   | ARC-c | ARC-e | Boolq | HellaSwag | MMLU  | OBQA  | PIQA  |  RTE  | WinoGrande | Average |
> > > > > | :---------------: | :--------: | :---: | :---: | :---: | :-------: | :---: | :---: | :---: | :---: | :--------: | :-----: |
> > > > > | Qwen1.5-MoE-A2.7B | Pretrained | 41.81 | 73.32 | 79.88 | 57.98 |  61.29 | 30.00 |  80.09 | 69.31|  68.98 |  62.58 |
> > > > > | Qwen1.5-MoE-A2.7B | SparseGPT (2:4, keep shared expert) | 33.62 | 67.05 | 71.01 | 43.87 | 42.29 | 26.00 | 74.10 | 62.45 | 65.51 |  53.98 |
> > > > > | Qwen1.5-MoE-A2.7B | Wanda (2:4, keep shared expert) | 30.29 | 62.12 | 64.59 | 40.68 | 37.63 | 23.40 | 72.14 | 57.40 | 64.48 | 50.30 |
> > > > > | Qwen1.5-MoE-A2.7B | NAEE (r=30, keep shared expert) | 32.25 | 59.34 | 67.28 | 46.74 | 38.08 | 21.20 | 73.50 | 64.26 | 60.46 | 51.46 |
> > > > > | Qwen1.5-MoE-A2.7B | MoE-Pruner (2:4, keep shared expert) | **39.93** | **71.21** | **71.53** | **52.73**  | **56.31** | **29.40** | **78.18** |  **70.04** | **67.80** | **59.68** |
> > > > >
> > > > >
> > > > > References:
> > > > >
> > > > > [1] Jeff Pool, Chong Yu. Channel Permutations for N:M Sparsity. NeurIPS 2021.
> > > > >
> > > > > [2] Yingtao Zhang, Haoli Bai, Haokun Lin, Jialin Zhao, Lu Hou, Carlo Vittorio Cannistraci. Plug-and-Play: An Efficient Post-training Pruning Method for Large Language Models. ICLR 2024.
> > > > >
> > > > > [3] An Yang, Baosong Yang, Binyuan Hui, Bo Zheng, Bowen Yu, Chang Zhou, Chengpeng Li, Chengyuan Li, Dayiheng Liu, Fei Huang, et al. Qwen2 technical report. arXiv preprint arXiv:2407.10671.

---

### Official Review · Reviewer_nNCF · 2024-11-04

**Soundness:** 2
**Presentation:** 2
**Contribution:** 1
**Rating:** 3
**Confidence:** 4

**Summary:**

The paper proposes a pruning method specifically designed for Mixture-of-Expert (MoE) models. It uses the Gate values from the MoE router to enhance weight importance evaluation during pruning. Additionally, it introduces an expert-wise approach for knowledge distillation to improve the performance of the pruned model.

**Strengths:**

1. New Insights: The paper provides valuable insights into how expert initialization methods influence final load balance and similarities between experts.

  2. Perplexity Performance Improvements: The proposed method leverages gate values in MoE to enhance the performance of pruned models, achieving perplexity improvements over previous methods designed for general transformer-based LLMs.

**Weaknesses:**

1. Efficiency Evaluation: The paper lacks an evaluation of the proposed method’s efficiency improvements, such as wall-clock speedup or memory reduction.

  2. Limited Technical Contribution: Pruning weights in the least-activated expert appears intuitive, yet it’s unclear how the method addresses potential drawbacks. For instance, which specific capabilities are impacted by this pruning? Does the improved performance on tested tasks come at the cost of decreased performance on other rare but crucial tasks?

  3. Unclear Contribution of Gate Value Insights: While the paper discusses gate value differences between two MoE initialization methods, it doesn’t clarify how these insights inform the pruning method’s design. Additionally, a brief definition of sparse upcycling initialization in related work would be helpful, as it is repeatedly referenced in later discussions.

  4. Writing and Support for Claims: Some technical claims lack proper references or justification. For example, the statement “MoE mitigates catastrophic forgetting in continual learning” (Line 44, Introduction) is not supported by references. Similarly, the claim that “expert activation frequency is task-agnostic” (Line 161, Methodology) seems conclusive, yet the paper does not provide corresponding experimental or logical evidence.

**Questions:**

1. What is the runtime efficiency of the proposed method?

  2. What is the maximum sparsity that the proposed method can achieve without incurring significant perplexity loss? Could you provide performance comparisons under moderate losses, such as a 10% perplexity increase compared to the dense model? The perplexity loss at 50% density appears a bit high for both the proposed method and baselines.

---

> ### Author Response · Authors · 2024-11-25
> **Response to Reviewer nNCF [1/2]**
>
> Dear reviewer nNCF,
>
> Thank you for taking the time to provide us with a thorough review and valuable suggestions. We are eager to respond to your questions and comments:
>
> > 1. Efficiency Evaluation: The paper lacks an evaluation of the proposed method’s efficiency improvements, such as wall-clock speedup or memory reduction.
>
> We conduct an **efficiency evaluation experiment** on memory reduction and inference speedup using the A100 GPU. To provide a comprehensive comparison, we also evaluated the NAEE [1] expert pruning method. Comparison results are shown in the table below. Our proposed MoE-Pruner method at the structured 2:4 sparsity pattern outperforms NAEE in terms of both average performance and inference speedup, while incurring only a small memory overhead for storing sparse tensor indices. The original average performance and memory of Mixtral-8x7B are 69.16 and 87.49GB, respectively. Speedup is measured according to the original Mixtral 8x7B baseline. The following table has been added to the Table 3 efficiency evaluation experiments in our revised version:
>
> | Model | Method | Sparsity | Average | Memory(GB) | Speedup |
> | ----------- | ----------- | ----------- | ----------- | ----------- | ----------- |
> | Mixtral-8x7B | NAEE[1] | r=4 | 61.69 | 45.49 | 1.06× |
> | Mixtral-8x7B | MoE-Pruner | 2:4 | 64.58 | 50.74 | 1.14× |
>
> > 2. Limited Technical Contribution: Pruning weights in the least-activated expert appears intuitive, yet it’s unclear how the method addresses potential drawbacks. For instance, which specific capabilities are impacted by this pruning? Does the improved performance on tested tasks come at the cost of decreased performance on other rare but crucial tasks?
>
>
> In our study, we tested pruned model performance across **9 zero-shot benchmarks**. Comparisons with weight pruning methods (SparseGPT and Wanda) and expert pruning method (NAEE [1]) show that MoE-Pruner achieves superior average performance. While all methods exhibit slight performance drops on individual benchmarks, MoE-Pruner preserves model capabilities better.
> - NAEE [1], for example, shows significant performance drops on **HellaSwag** and **MMLU**, indicating the loss of critical capabilities.
> - In contrast, MoE-Pruner exhibits **consistent performance across tasks**, balancing the trade-off between pruning and maintaining task diversity.
>
> > 3. Unclear Contribution of Gate Value Insights: While the paper discusses gate value differences between two MoE initialization methods, it doesn’t clarify how these insights inform the pruning method’s design. Additionally, a brief definition of sparse upcycling initialization in related work would be helpful, as it is repeatedly referenced in later discussions.
>
> Our method builds upon Wanda and extends it to MoE LLMs while maintaining the simplicity of Wanda's pruning metric. A key insight is the consideration of **router weights** ($Gate$) in MoE architectures. Consider a simple Mixture-of-Experts with two experts and each with only one weight: $y=Gate_1 \cdot w_1 \cdot x + Gate_2 \cdot w_2 \cdot x$. If $\vert w_1 \vert \leq \vert w_2 \vert$, traditional pruning methods would remove $w_1$. However, in MoE architectures:
> - If $Gate_1 \approx 1$ and $Gate_2 = 0$, $ \vert Gate_1 \cdot w_1 \cdot x \vert \gg \vert Gate_2 \cdot w_2 \cdot x \vert$, making $w_2$ the correct choice for removal.
>
> This motivating example shows that previous pruning methods for LLMs do not consider the importance of router weights which only exist in MoE architecture and may result in lower performance after pruning MoE. Therefore, we propose MoE-Pruner designed explicitly for MoE LLMs to handle such a limitation while maintaining the simplicity of Wanda's pruning metric.
>
> Thanks for your suggestions of adding the definition of sparse upcycling initialization.
> - To clarify sparse upcycling initialization: it starts from a dense checkpoint, with each new MoE expert layer initialized as a copy of the original MLP layer, excluding the MoE router [2].
> - This clarification has been added to Section 2, line 98, in the revised paper.
>
>
> > 4. Writing and Support for Claims: Some technical claims lack proper references or justification. For example, the statement “MoE mitigates catastrophic forgetting in continual learning” (Line 44, Introduction) is not supported by references. Similarly, the claim that “expert activation frequency is task-agnostic” (Line 161, Methodology) seems conclusive, yet the paper does not provide corresponding experimental or logical evidence.
>
> - For the statement "MoE mitigates catastrophic forgetting in continual learning" (Line 44, Introduction), we have added references [2] and [3] in our revised version.
> - We use samples from the pre-training dataset C4 as calibration data, since pre-training datasets are often more comprehensive and not dominated by knowledge specific to any particular domain [1].

---

> > ### Author Response · Authors · 2024-11-25
> > **Response to Reviewer nNCF [2/2]**
> >
> > > 1. What is the runtime efficiency of the proposed method?
> >
> > Please see our response to weakness 1, which includes detailed evaluations of memory reduction and inference speedup.
> >
> > > 2. What is the maximum sparsity that the proposed method can achieve without incurring significant perplexity loss? Could you provide performance comparisons under moderate losses, such as a 10% perplexity increase compared to the dense model? The perplexity loss at 50% density appears a bit high for both the proposed method and baselines.
> >
> > We add the pruning perplexity results comparisons under moderate losses. For a 10% perplexity increase, the unstructured pruning sparsity is usually about 40%. For structured pruning, we find that a 1:4 pruning pattern could have a moderate loss. We compare MoE-Pruner with SparseGPT and Wanda at the same level of pruning ratios and it clearly shows that MoE-Pruner outperforms SparseGPT and Wanda on all levels of pruning ratios:
> >
> > |           Model           |           Method            | Sparsity | Perplexity ↓ | Sparsity | Perplexity ↓ |
> > | :-----------------------: | :-------------------------: | :------: | :----------: | :------: | :----------: |
> > |       Mixtral-8x7B        |        Pretrained           |    -     |     3.84     |    -     |     3.84     |
> > |       Mixtral-8x7B        |      SparseGPT (40%)        |   40%    |     4.40     |   1:4    |     4.28     |
> > |       Mixtral-8x7B        |        Wanda (40%)          |   40%    |     4.33     |   1:4    |     4.25     |
> > |       Mixtral-8x7B        | MoE-Pruner (Ours, 40%)      |   40%    |  **4.20**    |   1:4    |  **4.14**    |
> > | Mixtral-8x7B-Instruct     |        Pretrained           |    -     |     4.14     |    -     |     4.14     |
> > | Mixtral-8x7B-Instruct     |      SparseGPT (40%)        |   40%    |     4.60     |   1:4    |     4.51     |
> > | Mixtral-8x7B-Instruct     |        Wanda (40%)          |   40%    |     4.57     |   1:4    |     4.49     |
> > | Mixtral-8x7B-Instruct     | MoE-Pruner (Ours, 40%)      |   40%    |  **4.48**    |   1:4    |  **4.41**    |
> >
> > We also update Table 2 perplexity results in our revised version.
> >
> > If you have other questions, we are happy to answer.
> >
> >
> > References:
> >
> > [1]. Xudong Lu, Qi Liu, Yuhui Xu, Aojun Zhou, Siyuan Huang, Bo Zhang, Junchi Yan, Hongsheng Li. Not All Experts are Equal: Efficient Expert Pruning and Skipping for Mixture-of-Experts Large Language Models. ACL 2024.
> >
> > [2] Aran Komatsuzaki, Joan Puigcerver, James Lee-Thorp, Carlos Riquelme Ruiz, Basil Mustafa, Joshua Ainslie, Yi Tay, Mostafa Dehghani, Neil Houlsby. Sparse Upcycling: Training Mixture-of-Experts from Dense Checkpoints. ICLR 2023.
> >
> > [3]. Mark Collier, Efi Kokiopoulou, Andrea Gesmundo, Jesse Berent. Routing networks with co-training for continual learning. arXiv:2009.04381.

---

> > ### Comment · Reviewer_nNCF · 2024-11-29
> >
> > Thank you for addressing my questions regarding reference clarity and experiments with reduced perplexity loss. However, the following points could further improve the paper:
> >
> > Response 1: The reported speedup (1.14x) seems minor given the significant performance drop (5%). Additionally, the benchmark used in the table is unclear. Is such a performance drop acceptable in this context?
> >
> > Response 3: I would still recommand the author to explicitly clarify how the observed "differences between two MoE initialization methods" incluence the pruning method's design. Otherwise, the two main contents of the paper seem a little disconnected in their current form.

---

> > > ### Author Response · Authors · 2024-12-03
> > > **Response to Reviewer nNCF [1/2]**
> > >
> > > Thank you for looking into our rebuttal and providing insightful suggestions to improve our paper quality. Here are our response to your followup questions:
> > >
> > > > Response 1: The reported speedup (1.14x) seems minor given the significant performance drop (5%). Additionally, the benchmark used in the table is unclear. Is such a performance drop acceptable in this context?
> > >
> > > The performance drop reported (from 69.16 to 64.58) reflects the results after one-shot structured pruning without fine-tuning. To address this, our paper proposes expert-wise knowledge distillation to recover pruned model performance using the pretrained MoE model with a limited compute budget. After applying this distillation (using 1,000 samples and training in less than one hour), the pruned model achieves a significantly improved average performance of 67.07, narrowing the gap to the original model.
> > >
> > > The 1.14× speedup was based on PyTorch 2.4.1, which only supports the CUTLASS backend for sparse tensor computations. With PyTorch 2.5.0 and the newly supported cuSPARSELt backend, the speedup improves to 1.31×, offering a more substantial inference efficiency gain. We provide the updated table reflecting these results:
> > >
> > > |  **Model**   | **Method** | **Sparsity** | **Average** | **Memory(GB)** | **Speedup** |
> > > | :----------: | :--------: | :----------: | :--------: | :--: | :--: |
> > > | Mixtral-8x7B | Pretrained | - | 69.16 | 87.49GB | 1.00× |
> > > | Mixtral-8x7B | NAEE | r=4 | 61.69 | 45.49GB | 1.06× |
> > > | Mixtral-8x7B | MoE-Pruner | 2:4 (CUTLASS) | 64.58 | 50.74GB | 1.14× |
> > > | Mixtral-8x7B | MoE-Distilled | 2:4 (CUTLASS) | 67.07 | 50.74GB | 1.14× |
> > > | Mixtral-8x7B | MoE-Pruner | 2:4 (cuSPARSELt) | 64.58 | 50.74GB | 1.31× |
> > > | Mixtral-8x7B | MoE-Distilled | 2:4 (cuSPARSELt) | 67.07 | 50.74GB | 1.31× |
> > >
> > > The benchmark is the same as the average zero-shot performance on 9 evaluation tasks we used in the paper: ARC-challenge, ARC-easy, Boolq, HellaSwag, MMLU, OpenBookQA (OBQA), PIQA, RTE, and WinoGrande. We also list the zero-shot performance below:
> > >
> > > |      Model       |      Method      |   ARC-c   |   ARC-e   |   Boolq   | HellaSwag |   MMLU    |   OBQA    |   PIQA    |    RTE    | WinoGrande |  Average  |
> > > | :--------------: | :--------------: | :-------: | :-------: | :-------: | :-------: | :-------: | :-------: | :-------: | :-------: | :--------: | :-------: |
> > > |   Mixtral-8x7B   |    Pretrained    |   56.91   |   84.47   |   85.29   |   64.78   |   67.03   |   35.00   |   82.43   |   70.40   |   76.16    |   69.16   |
> > > |   Mixtral-8x7B   |    NAEE (r=4)    |   48.38   |   77.99   |   80.52   |   57.81   |   47.68   |   28.60   |   78.67   |   62.45   |   73.16    |   61.69   |
> > > |   Mixtral-8x7B   | MoE-Pruner (2:4) |   47.87   |   79.00   |   79.54   |   58.86 |   62.17   |   31.80   |   79.49   |   68.23   |   74.27    |   64.58    |
> > > |   Mixtral-8x7B | MoE-Distilled (2:4) | **51.71** | **81.22** | **84.52** | **64.21** | **63.54** | **33.80** | **81.28** | **68.35** | **74.98** | **67.07** |
> > >
> > > To summarize, the performance drop after pruning (69.16 to 64.58) can be mitigated by our expert-wise knowledge distillation, achieving 67.07 performance. With additional data and compute resources, we anticipate further narrowing the gap to the pretrained model’s performance.

---

> > > ### Author Response · Authors · 2024-12-03
> > > **Response to Reviewer nNCF [2/2]**
> > >
> > > > Response 3: I would still recommend the author to explicitly clarify how the observed "differences between two MoE initialization methods" influence the pruning method's design. Otherwise, the two main contents of the paper seem a little disconnected in their current form.
> > >
> > >
> > > Thank you for this insightful observation. In our experiments, we observed that router weights (Gate values) differ significantly based on the MoE initialization method:
> > > - The router weights (Gate) in an upcycled model (e.g. Mixtral-8x7B) are not fully optimized in the MoE because they do not exist in the dense checkpoint and are only added after sparse upcycling.
> > > - The router weights in MoE model trained from scratch (e.g. DeepSeek-V2-Lite) are fully optimized during training, providing more reliable indicators of expert utilization and importance.
> > >
> > > By explicitly considering how the MoE initialization method affects the distribution and reliability of router and expert weights, we design a universal pruning metric that is adaptive to these differences:
> > > - For sparse upcycled models, the pruning method compensates for less reliable router weights by focusing on the inherent importance of expert weights and prevents the potential loss of valuable weights that might be mistakenly pruned due to under-optimized router weights.
> > > - For models trained from scratch, the pruning method takes full advantage of reliable router weights to make more informed pruning decisions and results in a more effective pruning process that maintains model performance by preserving critical experts and weights.
> > >
> > > We acknowledge that our paper did not sufficiently clarify this connection, leading to a disconnection between the MoE initialization methods and the pruning metric design. We will revise section 3.2 to explicitly discuss how MoE initialization methods impact router weight reliability and influence pruning design and add experimental results demonstrating how our pruning method performs on models initialized via both methods, highlighting the adaptations made for each case.

---

### Author Response · Authors · 2024-11-25
**General Response**

Dear reviewers,

We sincerely thank you for your insightful evaluations and constructive feedback! Your suggestions have greatly contributed to improving the quality of our paper. We address every question from each reviewer in the comments below. We summarize the key changes made to our revised manuscript, with changes highlighted in blue:

1. **Enhanced Literature Review:**
- We have moved the MoE initialization to section 2 for better organization.
- Additionally, we provided a brief definition of sparse upcycling initialization based on [1].
2. **Detailed Motivation and Analysis:**
- In Section 3.2, we expanded the motivation and analysis of our method, MoE-Pruner, to better explain its design and advantages over existing methods.
3. **Updated Experimental Results:**
- **Table 2**: We added results for pruning perplexity at lower sparsity levels, demonstrating that MoE-Pruner consistently outperforms SparseGPT and Wanda under moderate losses.
- **Table 3**: We introduced an efficiency evaluation table comparing memory reduction and inference speedup, including results for MoE-Pruner and the expert pruning baseline NAEE [2].
- **Table 4**: Updated zero-shot performance comparisons at **50% unstructured pruning**, confirming MoE-Pruner's robustness across tasks.
- **Table 5**: Added zero-shot performance comparisons at **structured 2:4 sparsity** and **50% expert pruning**, highlighting MoE-Pruner's scalability and superiority over SparseGPT, Wanda, and NAEE.
- These experimental results include **Mixtral-8x7B, MiniCPM-8x2B, DeepSeek-V2-Lite, and Qwen1.5-MoE-A2.7B** models.

These additions and updates comprehensively address your concerns and demonstrate **MoE-Pruner’s advantages in both performance and efficiency** across various pruning settings and architectures.

We hope the revised version reflects these improvements clearly. If you have other questions, we are happy to answer.

References:

[1] Aran Komatsuzaki, Joan Puigcerver, James Lee-Thorp, Carlos Riquelme Ruiz, Basil Mustafa, Joshua Ainslie, Yi Tay, Mostafa Dehghani, Neil Houlsby. Sparse Upcycling: Training Mixture-of-Experts from Dense Checkpoints. ICLR 2023.

[2]. Xudong Lu, Qi Liu, Yuhui Xu, Aojun Zhou, Siyuan Huang, Bo Zhang, Junchi Yan, Hongsheng Li. Not All Experts are Equal: Efficient Expert Pruning and Skipping for Mixture-of-Experts Large Language Models. ACL 2024.

---

### Note · Authors · 2024-12-16

I have read and agree with the venue's withdrawal policy on behalf of myself and my co-authors.